# THBS1-producing tumor-infiltrating monocyte-like cells contribute to immunosuppression and metastasis in colorectal cancer

**A list of authors and their affiliations appears at the end of the paper**

Mesenchymal activation, characterized by dense stromal infiltration of immune and mesenchymal cells, fuels the aggressiveness of colorectal cancers (CRC), driving progression and metastasis. Targetable molecules in the tumor microenvironment (TME) need to be identified to improve the outcome in CRC patients with this aggressive phenotype. This study reports a positive link between high thrombospondin-1 (THBS1) expression and mesenchymal characteristics, immunosuppression, and unfavorable CRC prognosis. Bone marrow-derived monocyte-like cells recruited by CXCL12 are the primary source of THBS1, which contributes to the development of metastasis by inducing cytotoxic T-cell exhaustion and impairing vascularization. Furthermore, in orthotopically generated CRC models in male mice, THBS1 loss in the TME renders tumors partially sensitive to immune checkpoint inhibitors and anti-cancer drugs. Our study establishes THBS1 as a potential biomarker for identifying mesenchymal CRC and as a critical suppressor of antitumor immunity that contributes to the progression of this malignancy with a poor prognosis.

Accumulating evidence has determined the critical role of stroma-infiltrating cells in the tumor microenvironment (TME) of colorectal cancer (CRC)[1–4]. CRC subtypes with mesenchymal phenotypes, consensus molecular subtype 4 (CMS4), exhibit aggressive behavior and poor prognosis with an immunosuppressive TME[5–7]. Although tumor-infiltrating myeloid cells constitute the primary population of immune components in the TME, targeting myeloid cells remains challenging because of the functional heterogeneity of their subsets[8]. Therefore, a deeper understanding of how myeloid cells contribute to the creation of a tumor-promoting microenvironment is essential for identifying rational combinatorial strategies against mesenchymal CRC.

Recent studies have revealed that CMS4 CRC have a higher expression of inflammation-related pathways, suggesting a strong association between inflammation and stromal activation[5,6,9,10].

Thrombospondin-1 (THBS1) is a matricellular protein that is highly expressed in inflammatory processes and has various functions, such as restraining angiogenesis and immune activity[11,12]. The anti-angiogenic effect of THBS1 is believed to contribute to tumor growth inhibition; however, the roles of THBS1 in carcinogenesis are multifaceted and contradictory[13–15]. To date, cell type-specific analyses of THBS1 function in intestinal tumorigenesis are lacking, and the contribution of THBS1 to CRC progression and metastasis remains unclear. Furthermore, the importance of THBS1 in regulating the immune landscape of TME remains to be uncovered.

In this work, we elucidate the cellular sources and functions of THBS1, particularly during the progression and metastasis of mesenchymal CRC, with a focus on the differences between the primary and metastatic sites. The combined effects of THBS1 loss

e-mail: yuki@kuhp.kyoto-u.ac.jp

and currently available chemotherapy and immune checkpoint inhibitors (ICI) are also determined. This study highlights the critical importance of bone marrow (BM)-derived monocyte lineages in enhancing tumor progression and provides evidence for the potential application of THBS1-targeted therapy as an effective treatment for aggressive CRC.

## Results

### Increased infiltration of THBS1-expressing cells in stroma correlates with aggressiveness of human CRC

Immunostaining of human CRC revealed that THBS1 was highly expressed in the stromal areas of both primary and metastatic lesions compared to the paracancerous tissues (Fig. 1a). The RNAscope for

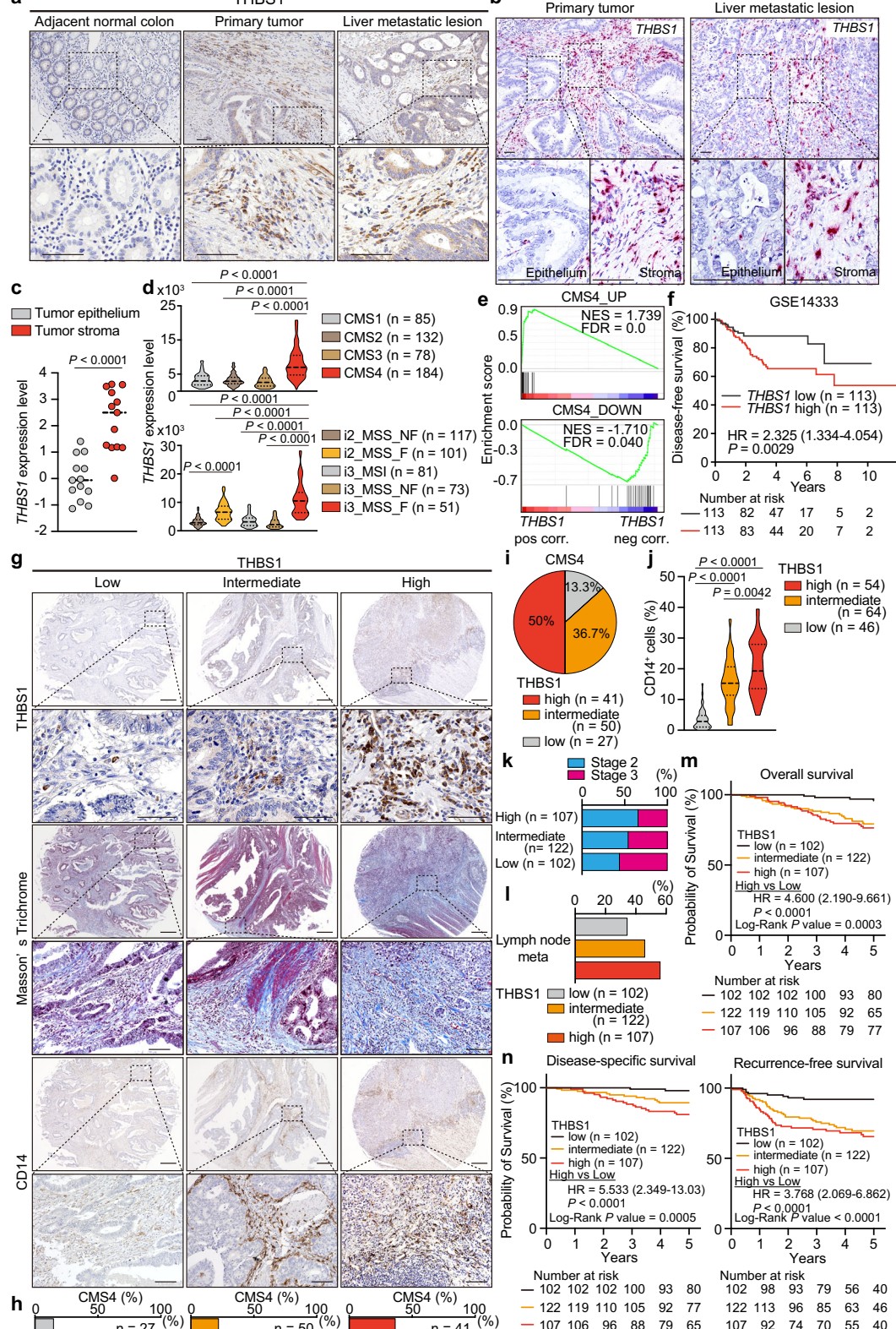

**Fig. 1 | THBS1 expression is primarily localized in the non-epithelial stroma areas of human CRCs and associated with the aggressiveness.**
**a**, **b** Representative images of immunostaining (**a**, *n* = 3 samples per indicated lesion) and RNAscope (**b**, *n* = 3 samples per indicated lesion) of THBS1 in human CRCs and liver metastasis. The experiments were independently repeated three times, yielding similar results. **c**, *THBS1* expression in tumor epithelium (*n* = 13), and tumor stroma (*n* = 13) of CRC dataset (GSE35602). **d** *THBS1* expression in each subtype of CMS (top) and IMF (bottom) in TCGA. **e** Gene set enrichment analysis (GSEA) of the indicated gene sets. **f** Kaplan-Meier curve in CRC dataset (GSE14333). **g**–**j** Staining **g** in TMAs of human CRCs (THBS1: low *n* = 102, intermediate *n* = 122,

high *n* = 107; the rest: low *n* = 46, intermediate *n* = 64, high *n* = 54). Proportion **h** of CMS4 subtype in indicated groups. Proportion **i** of CRCs with low-, intermediate- or high-intensity of THBS1 in CMS4-annotated CRCs. Quantification **j** of CD14 staining in **g**. **k**, **l** Proportion of stage 2 or 3 **k** and lymph node metastasis **l** in CRCs with low-, intermediate-, or high-intensity of THBS1 in TMAs. **m**, **n** Kaplan-Meier curves of patients corresponding to TMAs. Hazard ratio with 95% confidence interval and *P* values, analyzed by Log-rank test, are shown in **f**, **m**, **n**. Mean ± SEM. Scale bars, 50 μm **a**, **b**, top panels 200 μm and bottom panels 50 μm in each indicated protein **g**. *P* values were calculated by two tailed Mann-Whitney test in **c**, **d** or two-tailed, unpaired Student's *t* test in **j**. Source data are provided as a Source Data file.

THBS1 transcript and co-immunofluorescence of THBS1 with epithelial marker EPCAM confirmed the localization of THBS1 in the stroma, which was supported by transcriptomic analysis showing higher *THBS1* expression in the tumor stroma (Fig. 1b, c and Supplementary Fig. 1a). Analysis of the CRC dataset from the cancer genome atlas (TCGA) demonstrated a positive correlation between *THBS1* and mesenchyme-related genes (*VIM* and *FN1*) and pan-myeloid markers (*CD14* and *CD33*) (Supplementary Fig. 1b–e). THBS1 expression was highly enriched in mesenchymal CMS4 CRC and correlated with altered genes in this subtype (Fig. 1d, e). Recent advances in single-cell technology have identified distinct intrinsic epithelial signatures and defined the most aggressive fibrotic i3_MSS CRC (i3_MSS_F)[16], in which THBS1 expression was highly enriched (Fig. 1d). *THBS1*-high CRC exhibited an advanced stage, increased metastases to distant organs and lymph nodes, and a poorer prognosis, indicating the aggressive nature of these CRC (Fig. 1f and Supplementary Fig. 1f–i).

To further determine its clinical relevance, we applied tissue microarrays (TMAs) of surgically resected CRC[4]. High THBS1 expression was positively correlated with extensive stromal reaction, as indicated by dense collagen deposition, enrichment of CMS4 subtype, and accumulation of myeloid marker (CD14, CD11b, and CD68)-positive cells (Fig. 1g–j and Supplementary Fig. 1j, k). We found that the THBS1-high group correlated with a more advanced stage and increased lymph node metastases compared to the THBS1-low group (Fig. 1k, l and Supplementary Fig. 1l). Furthermore, high THBS1 expression was independently associated with recurrence, lymph node metastasis positivity, undifferentiated pathology, and microsatellite stable (MSS) status in the multivariate analyses (Supplementary Fig. 1m and Supplementary Tables 1–4). The THBS1-high group exhibited the lowest overall, disease-specific, and recurrence-free survival rates (Fig. 1m, n). These results highlighted the stromal localization of THBS1 and demonstrated a strong association between high THBS1 levels, aggressive phenotypes, and poor prognosis in human CRC.

**TME-derived THBS1 suppresses inflammation and immuno-surveillance in aggressive murine CRC**
To investigate the role of TME-derived THBS1 in aggressive CRC, we performed the orthotopic implantation of mouse tumor organoids (MTO) that had a CMS4-like mesenchymal phenotype, harboring mutations in *Apc*, *Trp53*, *Kras*, and *Tgfbr2* in the mouse rectum[17]. The implanted MTO led to the development of primary rectal lesions, followed by lymph node and liver metastases in immunocompetent mice, thereby successfully mimicking human CRC progression (Supplementary Fig. 2a). THBS1 expression was upregulated in primary and metastatic lesions in MTO-inoculated mice, like that in human CRC, and its stroma-specific localization was confirmed by RNAscope and immunofluorescence analyses (Supplementary Fig. 2b–d). MTO-derived tumors exhibited enhanced stromal activation, as indicated by abundant collagen deposition, accompanied by increased THBS1 expression, in contrast to tumors derived from MC38, a widely used mouse colon tumor cell line resembling immunogenic CRC (Supplementary Fig. 2e–g). This suggests that the high expression of THBS1 can be attributed to the mesenchymal phenotype of MTO-derived tumors.

We next analyzed orthotopic MTO tumors in *Thbs1*⁻/⁻ mice (Fig. 2a). This system enabled selective evaluation of the effect of THBS1 loss in non-epithelial cells of the TME. We observed a sharp reduction in THBS1 expression in tumors in *Thbs1*⁻/⁻ mice (Fig. 2b and Supplementary Fig. 2h, i), consistent with the stroma-specific enrichment of THBS1. Although there was no significant difference in the size of primary lesions between the genotypes, tumors in *Thbs1*⁻/⁻ mice displayed extensive necrotic areas (Fig. 2b and Supplementary Fig. 2i, j). Consistent with this, immunostaining revealed a substantial increase in cleaved caspase-3 (C-Cas3)⁺ apoptotic cells in tumors in *Thbs1*⁻/⁻ mice, which was reinforced by RNA sequencing (RNA-seq) analyses of these tumors, showing increased apoptotic pathways (Fig. 2b, c and Supplementary Fig. 2i, k). Consistent with the immunosuppressive function of THBS1[12], RNA-seq analyses of tumors in *Thbs1*⁻/⁻ mice also showed an upregulation of inflammatory and active immune responses (Fig. 2c). In addition, consistent with the anti-angiogenic properties of THBS1, tumors in *Thbs1*⁻/⁻ mice demonstrated increased CD31⁺ vascularity and angiogenesis-related signatures (Fig. 2d and Supplementary Fig. 2l, m). Collectively, these results highlight that THBS1 contributes to the development of an immunosuppressive and hypo-vascular TME.

**Loss of THBS1 blocks T cell exhaustion and increases stem-like CD8⁺ T cells**
To characterize the immune landscape governed by THBS1, we performed fluorescence-activated cell sorting (FACS) and immunohistochemical analyses of orthotopic MTO tumors. We observed increased infiltration of CD8⁺ T cells in tumors in *Thbs1*⁻/⁻ mice, although there was no significant change in CD4⁺ T cells or FOXP3⁺ regulatory T cells (Treg) (Fig. 2e, f and Supplementary Fig. 2n, o). Because THBS1 has been shown to have a potential regulatory effect on myeloid lineages[18], we examined myeloid cell populations including dendritic cells (DC), monocytes, neutrophils, and tumor-associated macrophages (TAMs) (Supplementary Fig. 2p); however, we observed no change in these populations. Myeloid-derived suppressor cells (MDSCs), including CD45⁺CD11b⁺CD11c⁻Ly6C⁺Ly6G⁻ monocytic MDSCs (MO-MDSCs) and CD45⁺CD11b⁺CD11c⁻Ly6C⁺Ly6G⁺ polymorphonuclear MDSCs (PMN-MDSCs), also did not show any changes (Supplementary Fig. 2q). Consistent with this, qRT-PCR confirmed that there was no significant change in the expression of myeloid-derived immunosuppressive genes such as *Il10*, *Arg1*, and *Mrc1* (Supplementary Fig. 2r). These results suggest the existence of a direct mechanism by which THBS1 regulates CD8⁺ T cell activity. To test this, we performed single-cell RNA-seq (scRNA-seq) on orthotopic MTO tumors. There were no major changes in the myeloid cell subsets and myeloid-related immunosuppressive gene expression such as M2 macrophage-related signature (Supplementary Fig. 3a–f). We further characterized the CD8⁺ T cell population in these tumors (Fig. 2g and Supplementary Fig. 3g). Analysis of orthotopic tumors in wild-type (WT) mice revealed that mT2_Exhausted CD8 cells were the most abundant population (Fig. 2g, h). In contrast, mT1_Stem-like CD8 cells were the most enriched in tumors in *Thbs1*⁻/⁻ mice, comprising more than 60% of all the infiltrating CD8⁺ T cells (Fig. 2g, h). Consistent with previous reports, mT1_Stem-like CD8 cells displayed an upregulation in genes related to active T cell

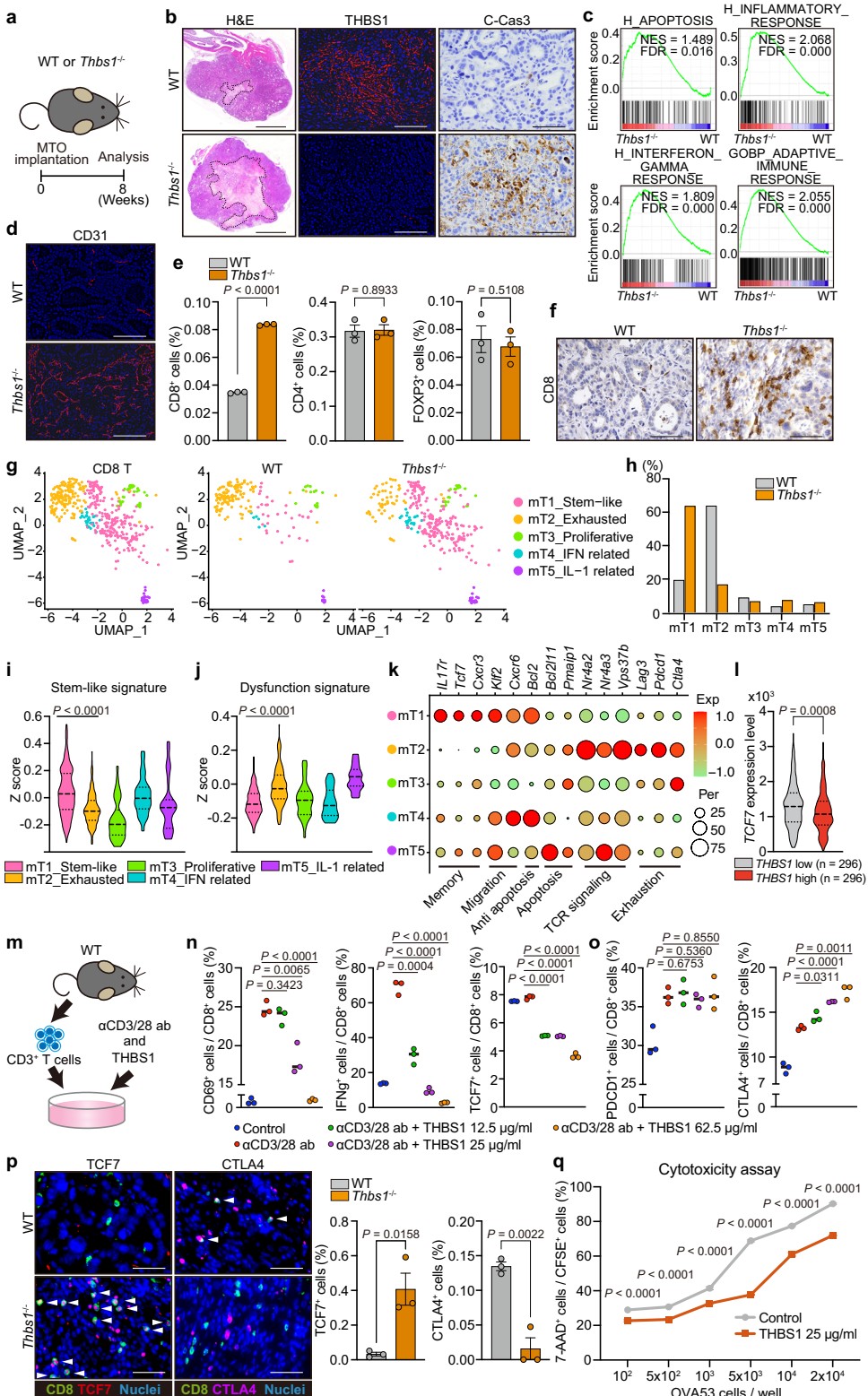

functions such as stemness, memory, migration, and anti-apoptosis and a downregulation of the dysfunctional signature (Fig. 2i–k)[19–21]. In contrast, mT2_Exhausted CD8 cells were enriched in genes related to the dysfunctional state, including apoptosis, T cell receptor (TCR) signaling, and exhaustion (Fig. 2j, k). Furthermore, FACS analysis demonstrated decreased CD8⁺ T cells expressing the dysfunctional marker CTLA4 in tumors in *Thbs1*⁻/⁻ mice (Supplementary Fig. 3h). These data highlighted the role of TME-derived THBS1 in inducing the

transition from an active stem-like state to dysfunction of CD8⁺ T cells. This is confirmed by the downregulation of TCF7, a stem-like T cell marker, in *THBS1*-high human CRC (Fig. 2l).

To examine the underlying mechanism by which THBS1 regulates T cell activity, we stimulated T cells isolated from the spleen with anti-CD3/CD28 antibodies in the presence of recombinant THBS1 (Fig. 2m). Consistent with bulk RNA-seq data, showing enhanced IFNγ signaling in tumors in *Thbs1*⁻/⁻ mice (Fig. 2c and Supplementary Fig. 3i), THBS1

**Fig. 2 | TME-derived THBS1 suppresses inflammation and immunosurveillance in aggressive murine CRC. a** Schematic representation of orthotopic MTO implantation. **b** Staining in orthotopic MTO tumors (*n* = 5 mice). Bars, 500 μm (H&E), 50 μm (the rest). Dash lines denote necrotic areas. **c** GSEA of transcriptomics of orthotopic MTO tumors in WT or *Thbs1*⁻/⁻ mice (*n* = 3). **d** Immunostaining in orthotopic MTO tumors (*n* = 5 mice per group). Bars, 50 μm. **e**, Proportion of indicated cells in total cells of orthotopic MTO tumors, analyzed by FACS (*n* = 3 tumors from three distinct mice per group). **f** Immunostaining in orthotopic MTO tumors (*n* = 5 mice per group). Bars, 50 μm. **g**–**k** scRNA-seq of orthotopic MTO tumors (*n* = 2 tumors from two distinct mice per group; analyzed cell numbers are indicated in Methods "Single-cell RNA sequencing"). UMAP plot **g** and proportion of each compartment **h** in re-clustered CD8⁺ T subset. Violin plots (**i, j**) and dot plots of representative genes **k** for the indicated gene signatures in each CD8 subset.

**l** Expression levels of *TCF7* in TCGA (*n* = 296 samples per group). **m** Schematic representation of stimulation experiment of isolated CD3⁺ T cells. **n, o** FACS analyses of proportion of indicated marker-expressing cells in CD8⁺ T cells in **m** (*n* = 3 biologically independent samples). **p** Co-immunostaining for CD8 and TCF7 (left) or CTLA4 (right) in primary tumors of MTO-inoculated mouse and proportion of double positively stained cells per field (*n* = 3 mice per group). White arrowheads denote double positive cells. Scale bars, 50 μm. **q** Cytotoxicity assay by co-culture of OVA-expressing MTO and OVA53 cells in the presence or absence of THBS1 (*n* = 3 biologically independent samples per group). Vertical axis indicates the proportion of dead cells (7-AAD⁺) in CFSE labeled OVA-expressing MTO. Staining experiments **b, d, f** were independently repeated three times. Mean ± SEM. Two-tailed, unpaired Student's *t* test in **e, n, o**–**q**, two-tailed Mann–Whitney test in **i, j, l**. Source data are provided as a Source Data file.

treatment suppressed anti-CD3/CD28-mediated T cell activation in a concentration-dependent manner as indicated by decreased proportions of CD69⁺, IFNγ⁺, and TCF7⁺ CD8⁺ T cells (Fig. 2n). Furthermore, THBS1 treatment increased the number of CTLA4⁺CD8⁺ T cells, although it did not significantly alter PDCD1⁺CD8⁺ T cells (Fig. 2o), which was consistent with the FACS results (Supplementary Fig. 3h). Immunostaining revealed an increase in TCF7⁺CD8⁺ T cells and a decrease in CTLA4⁺CD8⁺ T cells in tumors in *Thbs1*⁻/⁻ mice (Fig. 2p). These findings are consistent with a previous study that found suppressive activity of THBS1 on TCR-mediated T cell activation[22]. The strong suppression of ex vivo T cell activation by THBS1 seems at odds with the enrichment for exhausted/dysfunctional CD8⁺ T cells, which is generally associated with chronic TCR activation, in THBS1-high orthotopic MTO tumors. This discrepancy is probably attributed to the strong and acute activation of the ex vivo protocol compared to the more long-term and likely lower-level TCR triggering by endogenous antigens in vivo. To test whether THBS1 functionally attenuates T cell cytotoxicity, we performed a killing assay by co-culturing CFSE-labeled OVA-expressing MTO and OVA53 T cells in the presence of THBS1[23]. Notably, THBS1 treatment suppressed the cytotoxic activity of T cells under all conditions tested (Fig. 2q). The TMA analyses demonstrated a negative correlation between THBS1 expression and CD8⁺ T cell infiltration in human CRC (Supplementary Fig. 3j). These results demonstrate that TME-derived THBS1 contributes to immune evasion by directly suppressing T-cell activation and inducing dysfunction.

### Loss of TME-derived THBS1 suppresses metastasis

Given the role of cytotoxic T cells in preventing tumor progression, we investigated whether THBS1-deficiency in the TME suppresses the metastasis of mesenchymal CRC. MTO-bearing WT mice developed multiple metastases in the liver and lymph nodes (CK19⁺) (Fig. 3a, b and Supplementary Fig. 4a). In contrast, *Thbs1*⁻/⁻ mice inoculated with MTO demonstrated a sharp decrease in metastatic burden and prolonged overall survival (Fig. 3a–d, Supplementary Fig. 4a). We hypothesized that the increased infiltration of active CD8⁺ T cells may account for the reduction in metastasis in *Thbs1*⁻/⁻ mice. Anti-CD8 antibody treatment effectively depleted CD8⁺ T cells and concomitantly decreased the number of apoptotic cells in primary tumors (Fig. 3e, f), confirming the cytotoxic activity of infiltrating CD8⁺ T cells. Notably, CD8⁺ T cell depletion restored metastases in MTO-bearing *Thbs1*⁻/⁻ mice (Fig. 3g, h). To further clarify the impact of THBS1 loss on metastasis, we injected MTO into the spleen of WT and *Thbs1*⁻/⁻ mice and evaluated liver metastasis (Fig. 3i). Similar to the results of orthotopic implantation, *Thbs1*⁻/⁻ mice demonstrated decreased metastasis, which reverted to WT levels when CD8⁺ T cells were depleted (Fig. 3j, k). These data suggest that TME-derived THBS1 plays a pivotal role in enhancing the metastasis of intestinal tumors with a mesenchymal phenotype. Although immunostaining and transcriptomic analyses demonstrated the localization of THBS1 in the tumor mesenchyme, as described above, orthotopic tumors in *Thbs1*⁻/⁻ mice showed low levels of THBS1

(Supplementary Fig. 2h). However, we observed that *Thbs1*-knockdown MTO showed no significant difference in growth and metastatic formation in the splenic injection model (Supplementary Fig. 4b–g), suggesting no obvious contribution of tumor epithelium-derived THBS1 to tumor progression. Taken together, these findings indicate that TME-derived THBS1 plays a critical role in promoting CRC metastasis by restraining antitumor immunity.

### Higher antitumor immune activity in metastasis of *Thbs1*⁻/⁻ mice

Metastasis was strongly suppressed in *Thbs1*⁻/⁻ mice, whereas the primary tumor was not affected despite increased cell death in these mice, prompting us to compare the immune TME at primary and metastatic sites. We found no difference in the infiltration of THBS1-expressing cells between primary tumors and liver metastases in WT mice (Fig. 3l). However, increased infiltration of CD8⁺ T cells was more evident in metastasis in *Thbs1*⁻/⁻ mice (Fig. 3m and Supplementary Fig. 5a), suggesting a stronger induction of antitumor immunity in metastasis by THBS1-deficiency. FACS analyses of WT tumors demonstrated that most T lymphocytes, including CD8⁺ T cells, CD4⁺ T cells, and Tregs, and myeloid populations, including DCs, TAMs, monocytes, and neutrophils, were similar in both lesions (Fig. 3n, o). Notably, we observed a significantly lower number of PMN-MDSCs in metastases than in primary sites, whereas the number of MO-MDSCs was comparable between lesions (Fig. 3p). Because the low incidence and small size of metastasis in MTO-bearing *Thbs1*⁻/⁻ mice hampered sufficient sample collection from metastasis for FACS analyses, we performed immunofluorescence to determine the immune TME of metastasis in WT and *Thbs1*⁻/⁻ mice, which revealed no significant change in monocytic lineages and neutrophils (Supplementary Fig. 5b, c); however, the number of PMN-MDSCs in metastasis was significantly lower in both genotypes (Fig. 3q and Supplementary Fig. 5d). Based on these findings, we hypothesized that the change in the proportion of PMN-MDSCs might account for the difference in the degree of antitumor immunity between lesions. To directly examine the suppressive function of PMN-MDSCs and MO-MDSCs in WT and *Thbs1*⁻/⁻ tumors, we isolated MDSCs using a MACS-based MDSC isolation kit and co-cultured them with T cells stimulated with anti-CD3/CD28 antibodies (Supplementary Fig. 5e). Both PMN- and MO-MDSCs from WT mice showed reduced numbers of CD69⁺ active CD8⁺ T cells and increased numbers of CTLA4⁺ exhausted cells (Fig. 3r), confirming their immunosuppressive functions. Notably, while PMN-MDSCs from *Thbs1*⁻/⁻ mice exhibited no change in suppressive activity, this activity was abolished in MO-MDSCs from *Thbs1*⁻/⁻ mice (Fig. 3r), suggesting that THBS1 mainly contributes to the function of monocytic lineages. Together with the lower accumulation of PMN-MDSCs in metastasis, these results suggest that the THBS1-mediated immunosuppressive function of the monocytic lineage is more critical for metastasis formation and explain why the impact of THBS1 loss is more evident in metastasis. In support of this notion, an increase in the infiltration of active TCF7⁺CD8⁺ cells was more pronounced in metastases than in primary tumors of *Thbs1*⁻/⁻ mice (Fig. 3s and Supplementary Fig. 5f).

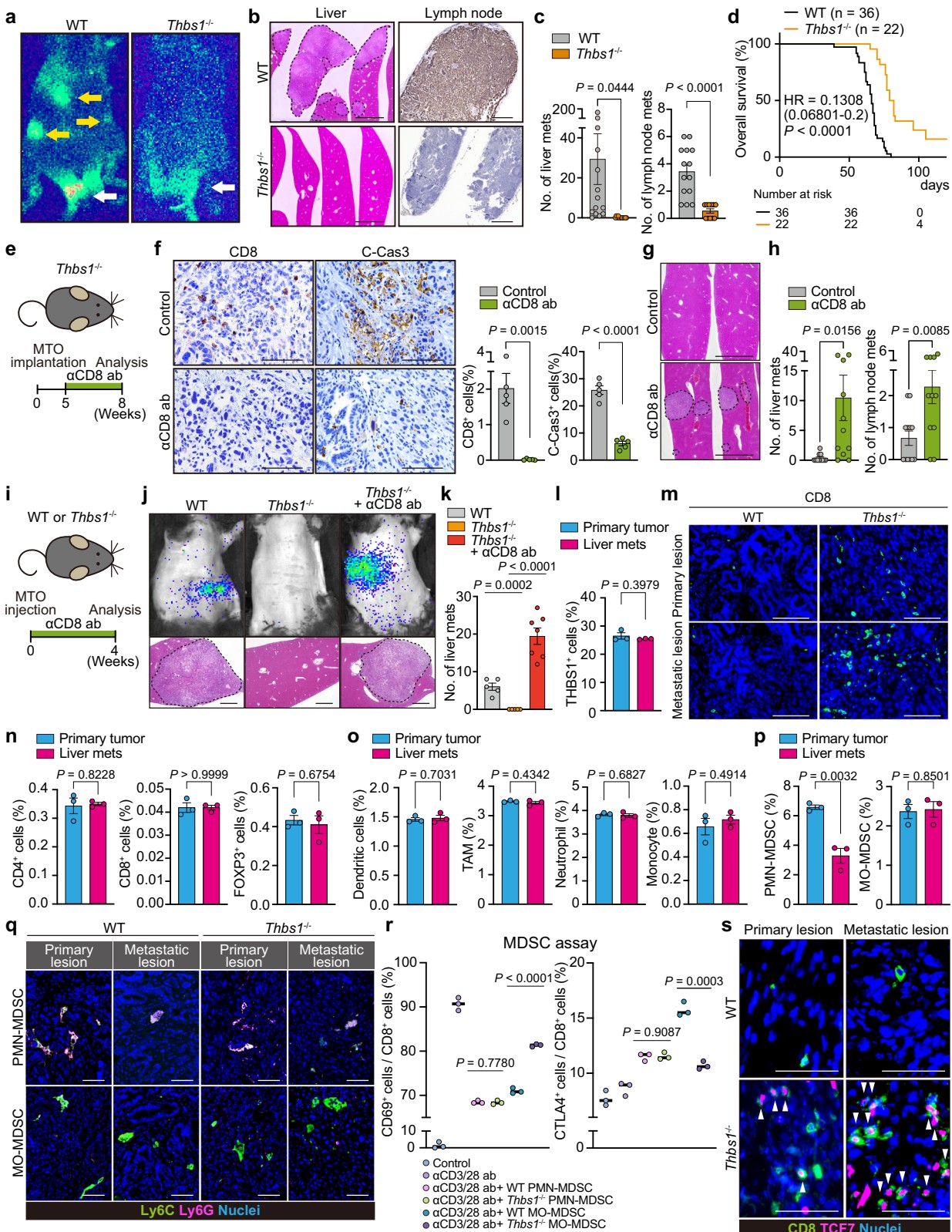

## THBS1-CD47 and THBS1-CD36 axes contribute to metastasis development

Although THBS1 activates latent TGFβ$_1$ in physiological state[22,24–28], analyses of RNA-seq data revealed an increased enrichment of TGFβ signaling in tumors in *Thbs1*[-/-] mice (Supplementary Fig. 6a, b). Furthermore, these tumors did not show a reduction in the levels of phospho-SMAD3, a representative downstream mediator of TGFβ

signaling, and cancer-associated fibroblast (CAF) marker αSMA (Supplementary Fig. 6c, d). To test whether THBS1 targets CD47 or CD36 to generate an immunosuppressive TME, we used *Cd47*[-/-] and *Cd36*[-/-] mice for orthotopic implantation of MTO (Fig. 4a). Tumors in *Cd47*[-/-] and *Cd36*[-/-] mice demonstrated increased CD8[+] cell infiltration and apoptosis compared to those in WT mice, although the extent of the increase was less than that in tumors in *Thbs1*[-/-] mice (Fig. 4b, c).

**Fig. 3 | Loss of TME-derived THBS1 suppresses metastases and improves prognosis. a–d** Orthotopic implantation of MTO. Bioluminescence imaging (**a**; white arrows: primary lesions, yellow arrows: distant metastases), H&E of liver (**b** left), CK19 staining of lymph node (**b**, right), macroscopic numbers of liver or lymph node metastasis (**c**; *n*: WT = 14, *Thbs1*[-/-] = 12), and Kaplan-Meier curve **d**. Hazard ratio with 95% confidence interval and *P* values, by Log-rank test. **e–h** Anti-CD8 antibody (αCD8 ab) treatment on MTO-inoculated *Thbs1*[-/-] mice (*n*: control = 12, αCD8 ab = 11). Schematic representation **e**, immunostaining and quantification in primary tumors in **e** (**f**, *n* = 5 mice per group), H&E of liver **g**, and macroscopic numbers of metastasis **h**. **i–k** αCD8 ab treatment on MTO splenic injection model (*n*: control = 5, *Thbs1*[-/-] = 7, *Thbs1*[-/-] + αCD8 ab = 7). Schematic representation **i**, bioluminescence imaging (**j**, top), H&E of liver (**j**, bottom), and macroscopic numbers of metastasis **k**. **l** FACS analyses of THBS1[+] cell proportion per total cells in indicated lesions in MTO-bearing WT mice (*n* = 3 mice per group).

**m** Immunostaining in indicated lesions. **n–p** FACS analyses on proportion of indicated cells among total cells in primary tumors or liver metastasis (*n* = 3 mice per group) in MTO-bearing WT mice. Dendritic cells, CD45[+]CD11c[+]; TAM, CD11b[+]F4/80[+]; neutrophils, CD45[+]CD11c[-]CD11b[+]Ly6C[+]Ly6G[+]F4/80[-]; monocytes, CD45[+]CD11c[-]CD11b[+]Ly6C[+]Ly6G[-]F4/80[-]; PMN-MDSC, CD45[+]CD11c[-]CD11b[+]Ly6C[+]Ly6G[+]; MO-MDSC, CD45[+]CD11c[-]CD11b[+]Ly6C[+]Ly6G[-] cells. **q** Co-immunostaining in indicated lesions. PMN-MDSC, Ly6C[+]Ly6G[+]; MO-MDSC, Ly6C[+]Ly6G[-] cells (*n* = 3 mice). **r** MDSC assay by co-culturing T cells isolated from WT mice and MO-MDSCs or PMN-MDSCs from primary tumors in MTO-bearing WT or *Thbs1*[-/-] mice, measured by FACS are shown (*n* = 3 biologically independent samples). **s** Co-immunostaining in indicated lesions (*n* = 3 mice). White arrowheads denote co-stained cells. Mean ± SEM. *P* values were calculated by two-tailed, unpaired Student's *t* test (except **d**). Scale bars, 500 μm (**b**, left; **g**); 200 μm (**b**,right; **j**). 50 μm **f**, **m**, **q**, 25 μm **s**. Dash lines denote liver metastases **b**, **g**, **j**. Source data are provided as a Source Data file.

Reflecting the difference in CD8[+] T cell infiltration, MTO-bearing *Cd47*[-/-] and *Cd36*[-/-] mice showed reduced liver metastases, although to a slightly lesser extent than MTO-bearing *Thbs1*[-/-] mice (Fig. 4d, e). Furthermore, reduced levels of lymph node metastases were observed in *Cd47*[-/-] mice, like those observed in *Thbs1*[-/-] mice, but not in *Cd36*[-/-] mice (Fig. 4e). Enhanced vascularization was observed only in tumors in *Cd36*[-/-] and *Thbs1*[-/-] mice (Fig. 4b, c), suggesting that the inhibition of angiogenesis was predominantly attributed to the action of THBS1-CD36. A previous report demonstrated that THBS1-CD47 signaling directly inhibits TCR-mediated T cell activation[22], while the THBS1-CD36 axis has been shown to reduce the inflammatory response by promoting IL10 secretion in myeloid cells[18]. However, IL10 levels were comparable between tumors in WT and *Cd36*[-/-] mice (Fig. 4f). A previous report has shown that CD36 signaling in intratumoral CD8[+] T cells attenuates the function of these cells[29]. Considering that THBS1 directly restricts CD3/CD28-mediated T cell activation (Fig. 2n, o), we hypothesized that both THBS1-CD47 and THBS1-CD36 inhibit anti-tumor immunity via a direct mechanism in CD8[+] T cells. Consistent with this notion, a substantial infiltration of CD47[+]CD8[+] and CD36[+]CD8[+] T cells was observed in tumors in WT mice (Fig. 4g). To examine the impact of CD47 or CD36 deficiency on CD8[+] T cells, we stimulated T cells isolated from *Cd47*[-/-] and *Cd36*[-/-] mice with anti-CD3/CD28 antibodies in the presence of THBS1 (Supplementary Fig. 6e, f). Notably, THBS1-mediated inhibitory effect was abolished in both CD47[-/-]CD8[+] T cells and CD36[-/-]CD8[+] T cells (Fig. 4h, i), suggesting that THBS1 suppresses T cell activity through binding with CD47 and CD36 on CD8[+] T cells. To validate the role of THBS1-CD36 in restricting vascularization, we performed a tube formation assay using mouse C166 endothelial cells. THBS1 dampened the tube-forming activity of C166, which reverted to normal levels when C166 was co-treated with sulfosuccinimidyl oleate, an irreversible inhibitor of CD36, but not with the anti-CD47 antibody (Fig. 4j). These data indicated the critical role of the THBS1-CD47 and THBS1-CD36 axes in establishing an immuno-suppressive microenvironment essential for tumor metastasis. Independent of its immunomodulatory function, THBS1 confers hypovascular characteristics to tumors by interacting with CD36.

### Tumor-infiltrating monocyte-like cells are primary producers of THBS1

Next, we aimed to determine the cellular source of THBS1 in the TME of CRC. Co-immunofluorescence revealed a strong colocalization of THBS1 and CD11b in orthotopic MTO tumors (Fig. 5a). To elucidate the role of THBS1 produced by myeloid cells, we used *LysM-Cre;Thbs1*[f/f] (*LysM;Thbs1*[Δ/Δ]) mice, in which *Thbs1* was selectively deleted from lysozyme M (LysM)-expressing myeloid cells for orthotopic implantation (Fig. 5b). Analysis of *LysM-Cre;Rosa26-LSL-EYFP* (*LysM;EYFP*) mice, in which LysM-expressing cells are labeled with yellow fluorescent protein (YFP), confirmed that THBS1 was expressed primarily by YFP[+] monocyte/macrophage lineages (Fig. 5c). Tumors in *LysM;Thbs1*[Δ/Δ] mice exhibited a substantial reduction in THBS1 expression (Fig. 5d

and Supplementary Fig. 7a). Notably, *LysM;Thbs1*[Δ/Δ] mice exhibited reduced metastatic burden and increased CD8[+] cell infiltration, vascularization, and apoptosis in primary tumors (Fig. 5e–g and Supplementary Fig. 7a, b), similar to *Thbs1*[-/-] mice.

To identify the myeloid subset responsible for THBS1 production, we analyzed the myeloid components in two independent scRNA-seq datasets of MTO tumors, using selective markers for each population (Fig. 5h and Supplementary Fig. 3c). Among these, monocyte-like cells expressing murine monocyte markers (*Ly6c2*, *Ccr2*, and *Vcan*) were the primary producers of THBS1 (Fig. 5h–j and Supplementary Figs. 3c, d, 7c). Analyses of the human CRC scRNA-seq dataset (SMC)[30] confirmed that monocyte-like cells expressing *CD14*, *FCN1*, *NLRP3*, and *VCAN* were the primary producers of THBS1 (Fig. 5k–m)[31]. A previous RNA velocity analysis identified a directional flow of CD14[+] monocytes toward myeloid lineages in tumors that express *FCN1* and *NLRP3*, suggesting that monocyte-like cells are a lineage migrating from the blood into tumors[31]. TCGA analysis also revealed that genes upregulated in monocyte-like cells were highly enriched in *THBS1*-high CRC and positively correlated with *THBS1* (Fig. 5m, n and Supplementary Fig. 7d). In support of the importance of monocyte-like cells in the progression of mesenchymal CRC, we observed an increased enrichment of monocyte-like cells and representative monocyte-like markers in CMS4 CRC (Fig. 5o, p). Consistent with these results, immunostaining exhibited strong colocalization of THBS1 and NLRP3 in CRC, further suggesting that monocyte-like cells are the source of THBS1 (Fig. 5q).

### THBS1-expressing monocyte-like cells are derived from BM

To demonstrate that THBS1-expressing monocyte-like cells were recruited from the BM, we compared the BM of MTO-bearing mice with that of untreated healthy mice and found increased expression of *Thbs1* and *Cd11b* in the BM cells of MTO-bearing mice (Fig. 6a). FACS analysis revealed an increase in the number of CD11b[+] cells in the BM and peripheral blood of MTO-bearing mice (Fig. 6b). Furthermore, CD45[+]CD11b[+] cells in the BM and peripheral blood demonstrated the highest *Thbs1* expression (Fig. 6c). Notably, patients with CRC exhibited higher levels of THBS1 in blood serum and primary CRC than those with non-cancerous lesions (Supplementary Fig. 8a, b). These results suggest that tumor formation at the primary site induces the BM to produce THBS1-expressing myeloid lineages that further migrate to the primary lesion. To demonstrate this, we transplanted BM from green fluorescent protein (GFP)-transgenic mice into WT mice and generated GFP-BM chimeric mice for MTO implantation (Supplementary Fig. 8c). EPCAM[-]GFP[+] BM-derived cells, that accounted for 21.2 ± 5.1% of the total cell population in tumors (Supplementary Fig. 8d), had the highest expression of *Thbs1* and *Vcan* compared to that in EPCAM[+] tumor epithelium or EPCAM[-]GFP[-] residential stromal cells and also compared to that in MTO or MC38 cells (Fig. 6d, e and Supplementary Fig. 8e). Furthermore, orthotopic tumors in GFP-BM chimeric mice showed strong co-localization of THBS1 and GFP

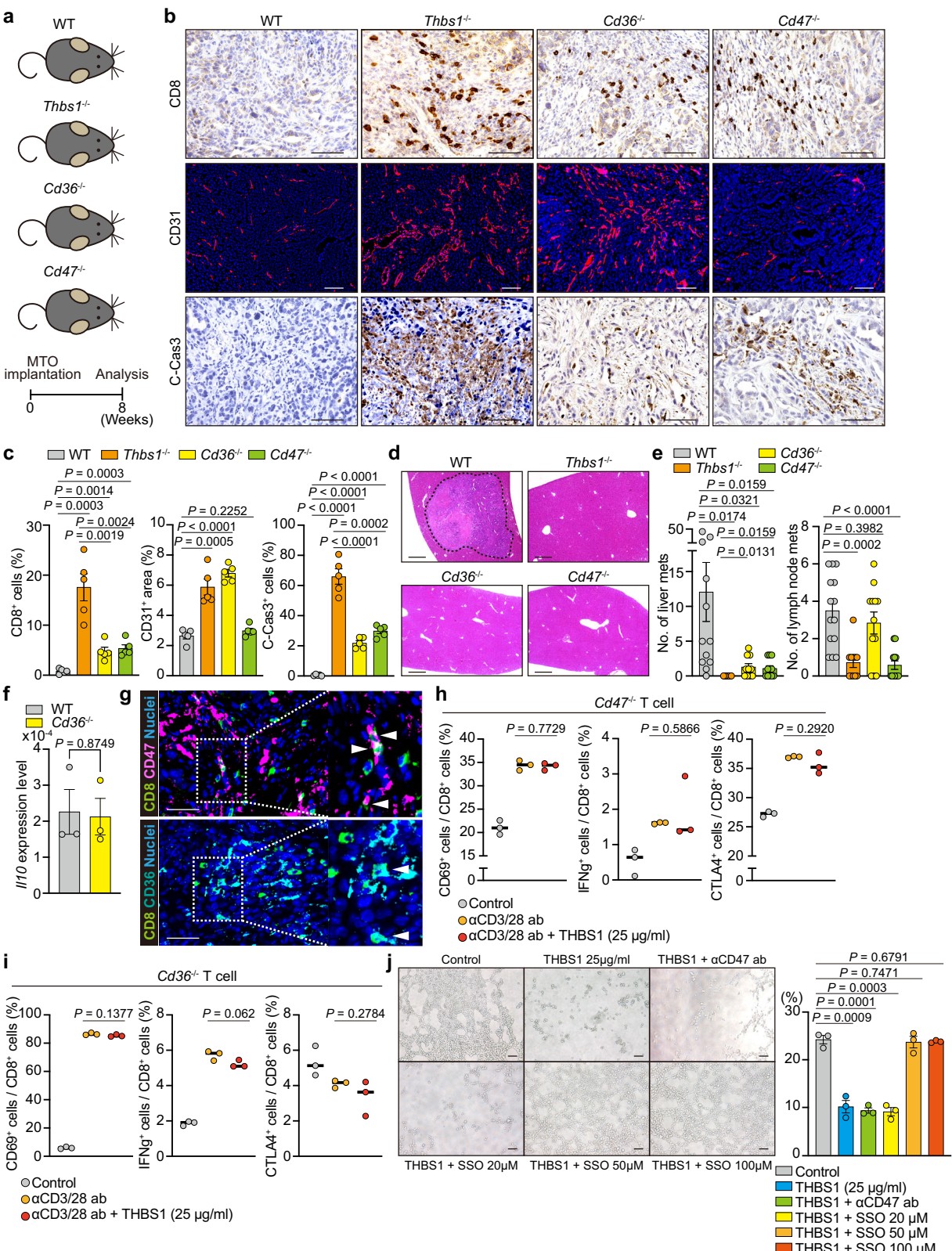

(Fig. 6f). To examine the impact of THBS1 restoration on the tumor-suppressive phenotypes of *Thbs1*[-/-] mice, WT-BM was transplanted into *Thbs1*[-/-] mice to generate WT-BM *Thbs1*[-/-] chimeric mice, followed by MTO implantation (Supplementary Fig. 8f). Notably, orthotopic tumors in WT-BM *Thbs1*[-/-] chimeric mice exhibited THBS1 levels equivalent to those in the WT control (Fig. 6g and Supplementary Fig. 8g). In addition, the reduced metastases and increased positivity

for CD31, CD8, and C-Cas3 observed in the primary tumors of *Thbs1*[-/-] mice reverted to WT levels after transplantation of WT-BM (Fig. 6h–j and Supplementary Fig. 8g). Together with the scRNA-seq results, these data confirmed that BM-derived monocyte-like cells are the primary producers of THBS1, which generates an immunosuppressive TME in mesenchymal tumors; this is also consistent with the contribution of THBS1 to the function of MO-MDSC (Fig. 3r).

**Fig. 4 | THBS1 enhances metastases through the interaction with both CD47 and CD36. a** Schematic representation of orthotopic implantation of MTO in the mice with indicated genotypes (*n*: WT = 14, *Thbs1*[-/-] = 11, *CD36*[-/-] = 12, *CD47*[-/-] = 12). **b–d** Immunostaining **b** and quantification of **b** (**c**; *n* = 5 mice per group) of the primary tumors and H&E staining **d** of liver of **a**. Dash lines denote liver metastases. **e** Macroscopic quantification of the numbers of liver and lymph node metastases in **a** (*n* = mice, per genotype: WT = 14, *Thbs1*[-/-] = 11, *CD36*[-/-] = 12, *CD47*[-/-] = 12). **f** qRT-PCR analysis in the orthotopic MTO tumors in WT and *Cd36*[-/-] mice (*n* = 3 mice per group). **g** Co-immunostaining for CD8 and CD47 (top) or CD36 (bottom) in primary lesions of MTO-bearing WT mice (*n* = 3 mice). White arrowheads denote co-stained cells. **h, i** FACS analyses of indicated cell proportions in CD8[+] (CD3[+]CD8[+]) T cells sorted from *CD47*[-/-] **h** or *CD36*[-/-] **i** mice. Individual plots indicate the biological replicates of isolated CD8[+] (CD3[+]CD8[+]) *CD47*[-/-] T cells or *CD36*[-/-] T cells (*n* = 3 per group) treated with or without anti-CD3/CD28 antibodies. **j** Representative images of tube formation assay of C166 cells in the presence or absence of recombinant THBS1 (25 µg/ml) or presence of THBS1 combined with anti-CD47 antibody (αCD47ab) or sulfosuccinimidyl oleate (SSO) at indicated concentration, and quantification of tube area (*n* = 3 biologically independent samples). Scale bars, 50 µm **b, g**, 100 µm **j**, 200 µm **d**. Mean ± SEM. *P* values were calculated by two-tailed, unpaired Student's *t* test. Source data are provided as a Source Data file.

## CXCL12-CXCR4 axis is critical for recruitment of THBS1-expressing cells

Aggressive CRC with a mesenchymal phenotype is characterized by a high density of CAFs, which release an array of chemokines depending on the epithelial subtype[6,16,32,33]. Given that BM cells in mice with mesenchymal MTO tumors demonstrated higher *Thbs1* expression than those in mice with immunogenic MC38 tumors (Fig. 6k), we sought to identify the CAF-derived factors that recruit THBS1-expressing cells from the BM. Similar to a previous report, TCGA analysis identified several chemokines specifically upregulated in CMS4 CRC (Fig. 6l and Supplementary Fig. 8h, i)[6]. Among them, CXCL12 and CCL2 are potent myeloid attractants expressed by CAFs[6,32,33]. *Cxcl12* expression was enriched in MTO-derived tumors, whereas that of *Ccl2* was enriched in MC38-derived tumors (Fig. 6m). In addition, *THBS1*-high human CRC showed prominent expression of *CXCL12* and its principal receptor, *CXCR4* (Fig. 6n). These data suggest that CXCL12 is responsible for the recruitment of THBS1-expressing monocyte-like cells. Consistent with this, we observed the highest *Thbs1* expression in CD45[+]CD11b[+]CXCR4[+] cells in the BM of MTO-bearing mice (Fig. 6o). Furthermore, consistent with the role of monocyte-like cells as primary producers of THBS1, FACS analyses identified CD45[+]CD11b[+]CXCR4[+]Ly6C[+]CCR2[+] cells as the dominant population for THBS1 expression in both primary and metastatic sites, which was supported by the colocalization of Ly6C, CXCR4, and THBS1 in tumors (Fig. 6p, q and Supplementary Fig. 8j). To further confirm this in vivo, MTO-bearing WT and GFP-BM chimeric mice were administered LIT-927, a selective neutraligand of CXCL12 (Fig. 6r). This treatment impaired the recruitment of GFP[+] cells and reduced THBS1 levels in the primary tumors of MTO-bearing GFP-BM chimeric mice (Fig. 6s and Supplementary Fig. 8k, l). Consistent with the results in *Thbs1*[-/-] mice, CXCL12 inhibition suppressed the metastatic burden in MTO-bearing mice with enhanced vascular formation, CD8[+] cell infiltration, and apoptosis (Fig. 6t, u and Supplementary Fig. 8k, l). Analysis of the SMC dataset revealed fibroblasts as the primary cluster expressing *CXCL12* in CRC (Supplementary Fig. 8m). Taken together, these results indicate that the primary lesion stimulates the BM to enhance the recruitment of THBS1-expressing cells via the action of CXCL12 produced by CAF, highlighting the interplay between CAF and myeloid signatures in mesenchymal CRC.

## THBS1 loss partially improves aggressive CRC response to current therapies

Currently, the beneficial effects of ICI are limited to microsatellite instability-high (MSI-H) CRC[5,34]. Consistent with the aggressive characteristics of THBS1-high CRC, analyses of TCGA and TMA revealed increased THBS1 expression in MSS CRC (Fig. 7a, b and Supplementary Fig. 1m). In contrast, anti-angiogenic therapy targeting vascular endothelial growth factor (VEGF) or its receptor (VEGFR) is an essential part of the current armamentarium. Given that the absence of THBS1 altered the TME from immunosuppressive to antitumor and induced hypervascularity (Fig. 2), we questioned whether these microenvironmental alterations induce therapeutic vulnerability to these treatments. To this end, *Thbs1*[-/-] mice were treated with an anti-PD1 antibody or anti-VEGFR2 antibody following MTO implantation (Fig. 7c). Both

treatments partially reduced primary tumor size with sustained reductions of liver and lymph node metastases in *Thbs1*[-/-] mice, but not in WT mice (Fig. 7c–h). ICI treatment enhanced CD8[+] T cell infiltration and cell death, whereas VEGF inhibition enhanced apoptosis and reduced angiogenesis in primary tumors of *Thbs1*[-/-] mice (Supplementary Fig. 9a–c).

Increased tomato lectin perfusion in tumors in *Thbs1*[-/-] mice suggested that the enhanced blood vessels induced by THBS1-deficiency were functional (Fig. 7i, j), implying the possibility of improved drug uptake. Therefore, we examined the combined effect of THBS1 inhibition and CRC chemotherapy, including folinic acid, fluorouracil, and oxaliplatin (FOLFOX) (Fig. 7k). FOLFOX treatment reduced primary tumor size in both WT and *Thbs1*[-/-] mice, but metastases were significantly suppressed in *Thbs1*[-/-] mice (Fig. 7k–p). Immunostaining revealed the most pronounced apoptosis in FOLFOX-treated *Thbs1*[-/-] mice (Supplementary Fig. 9d). Collectively, these results demonstrate that the loss of THBS1 results in a partially increased susceptibility of aggressive CRC to currently available treatments.

## Discussion

In this study, we demonstrated that THBS1 produced by BM-derived monocyte-like cells contributes to progression of aggressive intestinal tumors by suppressing cytotoxic immune activity, particularly at metastatic sites. Furthermore, we established a strong correlation between increased THBS1 expression and the mesenchymal and immunosuppressive features of human CRC with poor prognosis. Together with high THBS1 levels in the serum of CRC patients, our findings highlight the potential of THBS1 as a prognostic biomarker for CRC, particularly for the mesenchymal phenotype. Several reports have demonstrated elevated levels of THBS1 in other treatment-resistant cancers[35–38], suggesting that our findings in CRC may be applicable to a broad spectrum of malignancies with mesenchyme-/stroma-active characteristics.

A single-cell-based classification categorized mesenchymal CMS4/Fibrotic (F) CRC, previously assumed to be a coherent subtype, into i2_MSS_F and i3_MSS_F CRC. Notably, i3_MSS_F, which had the greatest propensity to metastasize, showed a stronger inflammatory response with increased monocyte/myeloid cell infiltration than i2_MSS_F[16]. However, in contrast to the MSI-H/CMS1 subtype, which is enriched with increased cytotoxic cells, the inflammation in i3_MSS_F illustrates a strong infiltration of immunosuppressive myeloid lineages, including monocytes, M2 TAM, and MDSC. THBS1 was mostly enriched in i3_MSS_F, suggesting that this molecule contributes to the formation of an immunosuppressive TME in this poor-prognosis subtype. The production of THBS1 by myeloid cells is critical for suppressing inflammation under certain pathological conditions[39,40]. Our analysis of the GFP-BM mice revealed a substantial number of infiltrating BM-derived myeloid cells in aggressive CRC. This is consistent with the fact that tumor cells activate the BM to produce an immature myeloid lineage with suppressive activity (MDSC), which plays a crucial role in cancer progression[41,42]. Combined with the results from *LysM-Cre* mice and scRNA-seq analyses, BM-derived monocyte/macrophage lineages were identified as the primary source of THBS1 in CRC. TAM have been suggested to originate from resident tissue macrophages or

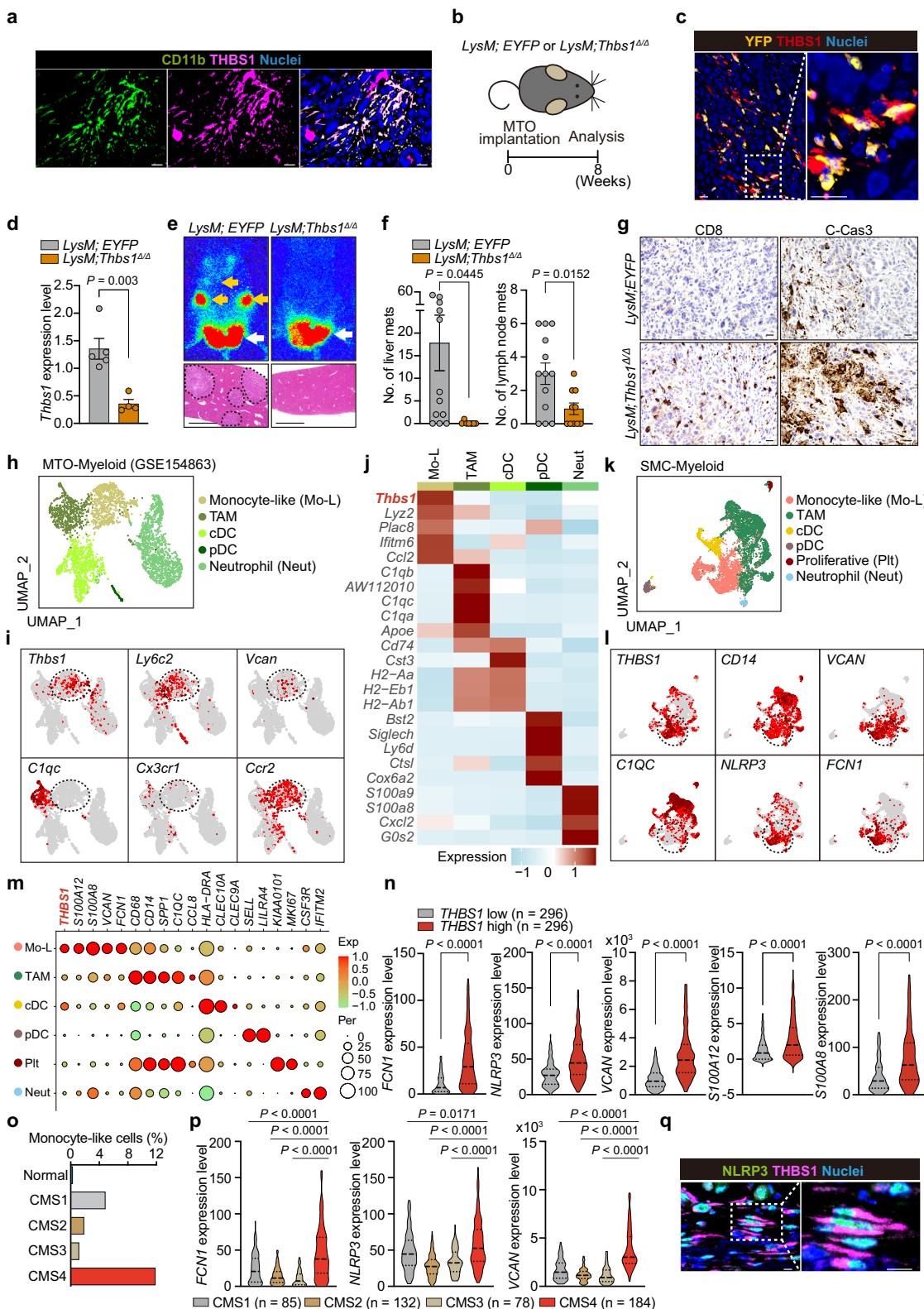

newly recruited monocytes via MO-MDSC, although it remains challenging to precisely distinguish between monocytes, MO-MDSC, and TAM in their differentiation and functional status[43,44]. In contrast, recent single-cell transcriptional trajectory analyses of myeloid lineages identified a strong directional flow of differentiation from monocytes toward FCN1+ monocyte-like cells, which subsequently gave rise to SPP1+ TAM[31]. The fact that mesenchymal CRC is infiltrated by THBS1-expressing monocyte-like cells that suppress antitumor immunity fills

a gap in our understanding of the recruitment and action of monocytic lineages to regulate the immune TME before they mature into TAM.

Intriguingly, THBS1 loss induced distinct tumor and immune phenotypes in primary and metastatic lesions. A comparison of the immune TME revealed a lower accumulation of PMN-MDSCs in metastases, whereas MO-MDSCs infiltrated at the same levels in both lesions. This suggests that THBS1-expressing monocytic lineages are critical modulators of metastasis, consistent with the notion that MO-

**Fig. 5 | Monocyte-like cells are the primary source of THBS1.**
**a** Co-immunostaining of the orthotopic MTO tumors in WT mice (*n* = 3 mice).
**b** Schematic representation of orthotopic implantation of MTO to the mice with indicated genotypes. **c** Co-immunostaining in the MTO tumors in *LysM;EYFP* mice (*n* = 3 mice). **d** qRT-PCR analysis in the orthotopic MTO tumors in *LysM;EYFP* (*n* = 5 mice) and *LysM;Thbs1*$^{\Delta/\Delta}$ mice (*n* = 4 mice). **e** Bioluminescence imaging (top) and H&E of liver (bottom) of **b**. **f** Macroscopic numbers of metastases in liver (*n* = mice per group: *LysM;EYFP* = 12, *LysM;Thbs1*$^{\Delta/\Delta}$ = 7) and lymph node (*n* = mice per group: *LysM;EYFP* = 13, *LysM;Thbs1*$^{\Delta/\Delta}$ = 10) of **b**. **g** Immunostaining in the orthotopic MTO tumors of **b** (*n* = 5 mice). **h–j** UMAP plot **h** of murine myeloid clusters, and UMAP plots **i** and heatmap **j** of representative genes for each cluster of (**h**) in scRNAseq data of orthotopic MTO tumors (GSE154863). Dash lines denote monocyte-like cluster. **k–m** UMAP plot **k** of human myeloid clusters and UMAP plots **l** and dot plots **m** of representative genes for each cluster of **k** in SMC dataset (GSE132465). Dash lines denote monocyte-like cluster. **n** Transcript levels of indicated genes in TCGA (*n* = 296). **o** Proportions of monocyte-like cells in myeloid cellular compartment of SMC, stratified by CMS subtypes. **p** Violin plots for indicated genes in TCGA, stratified by CMS subtypes (n: CMS1 = 85, CNS2 = 132, CMS3 = 78, CMS4 = 184). **q** Co-immunostaining in human CRC (n = 3 samples). Arrows in **e**: primary lesion (white), distant metastases (yellow). Dash lines denote liver metastases in **e**. Scale bars, 10 μm (**a, c, g, q**), 500 μm **e**. Immunostaining experiments (**a, c, g, q**) were independently repeated at least three times, yielding similar results. *P* values were calculated by two-tailed, unpaired Student's *t* test in **d, f**, or two-tailed Mann–Whitney test **n, p**. Source data are provided as a Source Data file.

MDSC creates a premetastatic niche[42]. Notably, THBS1 loss hindered the immunosuppressive function of MO-MDSCs but not of PMN-MDSCs, which agrees with the fact that monocyte-like cells, defined as Ly6c$^+$ BM-derived cells, were the primary producers of THBS1. Our findings provide insights into the mechanism by which MO-MDSCs promote tumor progression via THBS1 production, particularly during metastasis. Furthermore, this is consistent with the fact that a combination of THBS1 inhibition and ICI or chemotherapy showed a significant but only partial effect on primary lesions that equally comprised MO- and PMN-MDSCs. Therefore, therapeutics targeting PMN-MDSCs are required to enhance the antitumor effects of THBS1 inhibition on primary lesions.

Another clinically relevant effect of THBS1 loss is increased vascularity. The THBS1-CD36 axis strongly suppresses angiogenesis by inhibiting the proliferation and survival of endothelial cells[24,45,46]. This is consistent with our finding that orthotopic tumors in *Cd36*$^{-/-}$ mice, but not in *Cd47*$^{-/-}$ mice, showed increased vascularity. Several clinical trials applying THBS1-mimetics or analogs to enhance the inhibitory activity of THBS1 on vascularization failed to demonstrate beneficial efficacy in metastatic melanoma, advanced renal cell carcinoma, and soft tissue sarcoma[13–15], suggesting that reduced vascularity induced by THBS1 may play a key role in creating a therapy-resistant microenvironment in aggressive tumors.

In summary, our study contributes to a better understanding of the processes underlying THBS1 production by monocyte-like cells that mediate immune evasion and hypovascularity and confer metastatic and therapy-resistant features to CRC. These findings provide insights for the detection and treatment of this poor-prognosis malignancy.

## Methods

### Human samples and ethics approval
Paraffin-embedded tissue sections of CRC and adenoma were obtained from surgically resected CRCs and endoscopically resected adenomas. Serum samples were obtained from 37 patients (sex, male: *n* = 19, female: *n* = 18; age, 36–91 (median = 72); Stage, 0: *n* = 5, 1: *n* = 12, 2: *n* = 8, 3: *n* = 12) with CRCs and 19 patients with benign tumors (sex, male: *n* = 15, female: *n* = 4; age, 57–89 (median = 72); histology, adenoma: *n* = 15, sessile serrated lesion: *n* = 4). None of the patients had undergone preoperative radiation or chemotherapy. Written informed consent was obtained from above patients with the protocol approved by the Ethics Committee of Kyoto University Graduate School and Faculty of Medicine. TMAs of 331 surgically resected human CRC samples (sex, male: *n* = 169, female: *n* = 162; age: 21–96 (median = 68)) at Osaka Metropolitan University Hospital were applied for survival analyses, univariable and multivariable analyses. The Ethics Committee of Kyoto University and Osaka Metropolitan University approved the use of patient TMA samples for this experiment without requiring written informed consent. Informed consent was obtained in the form of opt-out on the website. Baseline characteristics are shown in Supplementary Fig. 1l. Participant compensation was not provided for both cohorts. Among these TMA samples, 118 CRCs were previously annotated to CMS4 (*n* = 30) or non-CMS4 (CMS1–3) by immunostaining-based CMS classifier[4]. CRC samples in TMA were classified into THBS1-low (*n* = 102), -intermediate (*n* = 122) and -high (*n* = 107) according to staining intensity of THBS1. To evaluate fibrosis, Masson's trichrome staining (Sigma-Aldrich) was performed. CD14, CD11b, CD68 and CD8 were stained in 164 CRC samples in the TMA. Opal multiplex IHC (NEL810001KT; Akoya) was performed to assess the loss of the mismatch repair (MMR) proteins, MLH1, PMS2, MSH2, and MSH6 in TMA (*n* = 301) using goat anti-mouse IgG1 (1:500, PA1-74421; Thermo Fisher) as a secondary antibody. Non-detection of nuclear staining of any of the MMR proteins in the tumor cells was interpreted as MMR-deficient (MSI-H) (MSI-H, *n* = 97; MSS, *n* = 204). The adjacent stromal cells were used as internal controls. The following primary antibodies were used for immunohistochemical analyses: THBS1 (1:500, MA5-13398; Thermo Fisher), CD14 (1:500, 75181; Cell Signaling), CD11b (1:1000, ab133357; Abcam), CD68 (1:500, 76437; Cell Signaling), CD8 (1:200, 85336; Cell Signaling), EPCAM (1:100, ab71916; Abcam) and NLRP3 (1:100, NBP2-12446; Novus Bio), MLH1 (1:25, 47954; Cell Signaling), PMS2 (1:100, 556415; BD Biosciences), MSH2 (1:100, NA27; Calbiochem), MSH6 (1:100, 610919; BD Biosciences). RNAscope for THBS1 (probe no. 426581; ACD) were performed in primary tumor site and metastatic liver tumor site of human CRC sample. All images were collected from each stained slide on a BZ-X710 microscope (KEYENCE). The number of whole cells and positively stained cells in IHC images were measured using the QuPath v.0.3.2[47], and the whole area and positively stained area in IF images were measured by ImageJ/Fiji software v.2.3.0/1.53 f[48].

### Mice
*Thbs1*-deficient (*Thbs1*$^{-/-}$ (#006141)), *LysM-Cre* (#004781) and *Rosa26-EYFP* (#006148) mice were purchased from Jackson Laboratories. C57BL/6 mice, GFP transgenic mice (C57BL/6-Tg (*CAG-EGFP*)) mice were purchased from Japan SLC,Inc. Systemic CD47-deficient (*CD47*$^{-/-}$) (*CD47*$^{f/f}$;*E2A-Cre*) mice were generated by crossing *CD47*$^{f/f}$ mice with *E2A-Cre* mice[49]. *Thbs1*$^{f/f}$ mice were generated as previously described[39]. CD36-deficient (*CD36*$^{-/-}$) mice were kindly provided by Dr. Freeman (Harvard Medical School). All mouse strains were generated in a C57BL/6 background and were born and maintained under pathogen-free conditions. These mice were maintained in 14 hr light/10 h dark cycle, and the housing temperature and humidity were maintained were 24 °C and 50%, respectively. Animal handling and experimental procedures conformed to institutional guidelines and were approved by the animal research committee of Kyoto University (Kyoto, Japan) and performed in accordance with Japanese government regulations. 8–10 weeks old male mice were used in all experiments.

### Generation of bone marrow (BM) chimeric mice
For the generation of GFP-BM chimeric mice, GFP transgenic mice were euthanized, and their BM cells were collected from their femur and humerus. Collected BM cells were admixed with the Red Blood Cell Lysis Solution (130-094-183; Miltenyi Biotec) for depletion of red blood cells, and diluted in HBSS. Recipient mice (C57BL/6 or *Thbs1*$^{-/-}$)

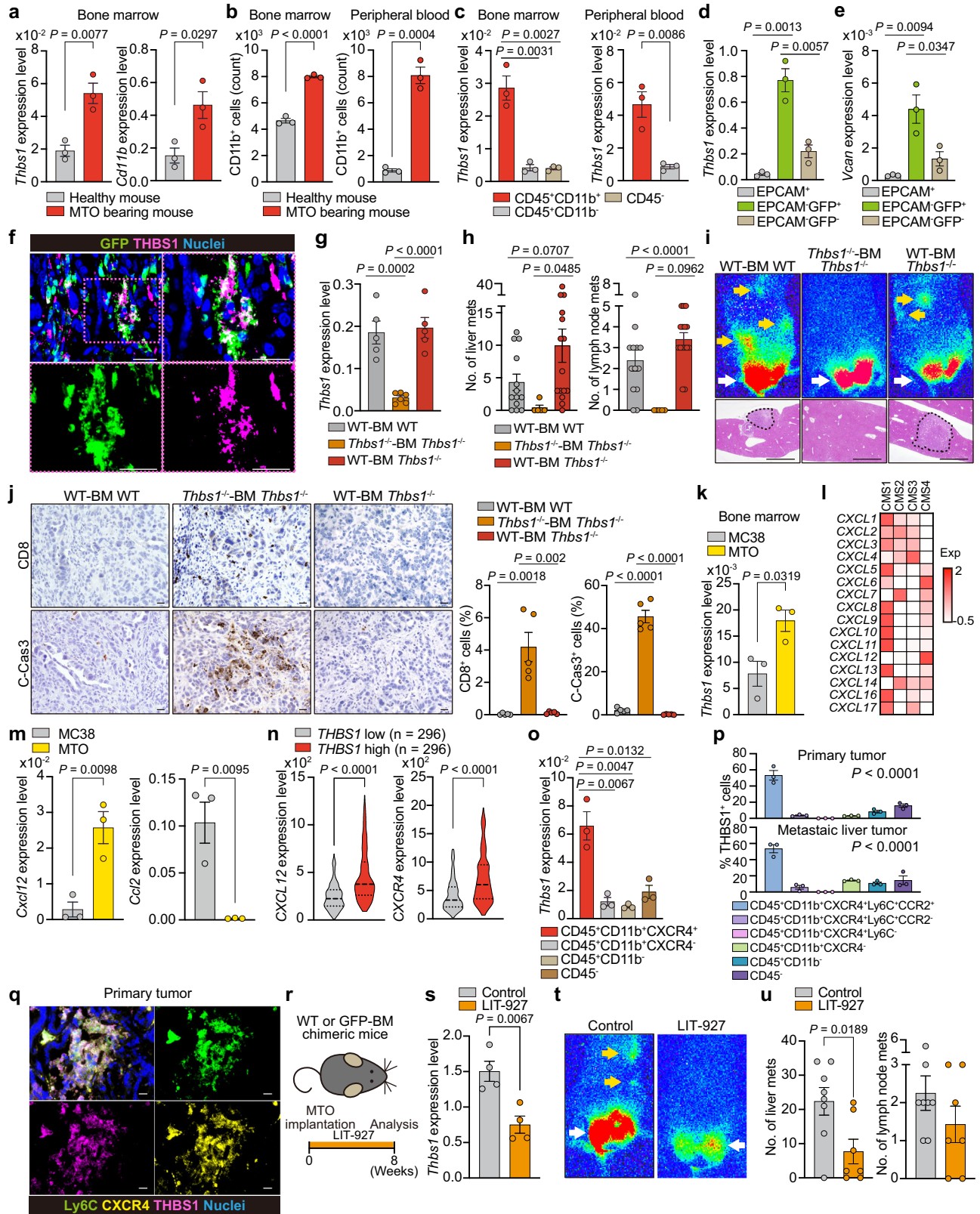

were exposed to 10 Gy total body irradiation (TBI) and intravenously injected the collected donor BM cells via tail vein. For the generation of WT-BM *Thbs1*[-/-] chimeric mice, BM cells were collected from C57BL/6 mice and transplanted to *Thbs1*[-/-] mice. The recipient mice were bred with 2 mg/ml neomycin-supplemented drinking water for 6 weeks after BM transplantation, followed by orthotopic implantation.

## Cell and organoid culture

Mouse tumor organoids (MTO: kindly provided by Dr. Batlle's laboratory) were cultured in advanced DMEM supplemented with 1% penicillin-streptomycin, Glutamax (gibco), N-2 Supplement (gibco) and B-27 Supplement (gibco)[17]. MC38 cells (kindly provided by Dr. Honjo's laboratory) were cultured in DMEM supplemented with 10% FBS and 1% penicillin-streptomycin. C166 cells were purchased from

**Fig. 6 | BM-derived THBS1-expressing cells are recruited by CXCL12 signaling.**
**a** qRT-PCR analyses in BM cells (*n* = 3 mice). **b** FACS analyses in the BM and peripheral blood cells (*n* = 3 mice). Positive cell numbers in 10000 total cells are shown. **c** qRT-PCR analyses of sorted cells from BM or peripheral blood of MTO-bearing mice (*n* = 3 mice). **d, e** qRT-PCR analyses of sorted cells from MTO tumors in GFP-BM mice (*n* = 3 mice). **f** Co-immunostaining of the MTO tumors in GFP-BM mice (*n* = 3 mice). **g** qRT-PCR analyses of the MTO tumors in WT-BM WT (*n* = 5), *Thbs1*⁻/⁻-BM *Thbs1*⁻/⁻ (*n* = 6), and WT-BM *Thbs1*⁻/⁻ chimeric mice (*n* = 5). **h** Macroscopic numbers of metastasis (n: WT-BM WT = 13, *Thbs1*⁻/⁻-BM *Thbs1*⁻/⁻ = 5, WT-BM *Thbs1*⁻/⁻ chimeric mice = 15). **i** Bioluminescence imaging (top) and H&E staining of liver (bottom) in indicated mice with MTO implantation. **j** Immunostaining in MTO tumors and positive cell proportion (*n* = 5 mice). **k** qRT-PCR analyses in BM cells of MC38- or MTO-bearing mice (*n* = 3 mice). **l** Heatmap in TCGA (n: CMS1 = 85,

CNS2 = 132, CMS3 = 78, CMS4 = 184). **m** qRT-PCR analyses in indicated orthotopic tumors in WT mice (*n* = 3 mice). **n** Transcript levels in TCGA. **o** qRT-PCR analyses of sorted cells from BM of MTO-bearing mice (*n* = 3). **p** FACS analyses of proportions among THBS1⁺ cells in MTO-bearing WT mice (*n* = 3). **q** Co-immunostaining in WT tumors (*n* = 3 mice). **r–u** CXCL12 inhibitor (LIT-927) treatment, following orthotopic MTO implantation (n: control = 8, LIT-927 = 7). Schematic representation (**r**), qRT-PCR analyses (**s**, *n* = 4), bioluminescence imaging **t**, and quantification of macroscopic numbers of metastases **u**. Arrows in **i, t**: primary lesions (white), distant metastases (yellow). Dash lines denote liver metastases in **i**. Scale bars, 10 μm **f, j, q**, 500 μm **i**. Mean ± SEM. Two tailed, Mann–Whitney test **n**, one-way ANOVA **p**, or tow tailed, unpaired Student's *t* test (the rest). Source data are provided as a Source Data file.

---

ATCC (CRL-2581) and cultured in DMEM supplemented with 10% FBS and 1% penicillin-streptomycin. OVA53 cells (kindly provided by Dr. Takehito Sato) were cultured in RPMI supplemented with 10% FBS and 1% penicillin-streptomycin[23]. Isolated CD3⁺ T cells and PMN-/MO-MDSC were cultured in RPMI supplemented with 10% FBS and 1% penicillin-streptomycin.

### Orthotopic implantation of organoids to murine rectum
Eight to ten weeks old mice were anaesthetized by isoflurane inhalation and placed in a supine position. Rectal mucosae were exposed by insertion of two blunt-ended hemostats into the anus. A 20 μl cell suspension of 10⁵ cells of MTO or MC38 admixed with matrigel (Corning) was directly injected into the colonic mucosae. Mice implanted with MTOs were euthanized 8–9 weeks after injection. Mice implanted with MC38 were euthanized 5 weeks after injection because the MC38-bearing mice died at around 5 weeks after implantation. Orthotopic primary lesions, liver and lymph node metastases were scored macroscopically as well as histologically. The maximal tumor burden permitted by the ethics committee was 10% of body weight. We ensured that each time mice were sacrificed, the maximal tumor burden did not exceed this limit.

### Splenic injection of organoids
Eight to ten weeks old mice were anaesthetized by isoflurane inhalation and placed in a right lateral decubitus position. A 100 ml cell suspension of 10⁵ cells of MTO was injected into spleen, followed by partial splenectomy of injected lesion. Mice were euthanized 4 weeks after injection and liver metastasis were scored macroscopically.

### Bioluminescence imaging
Bioluminescence imaging was used to track luciferase expression of orthotopically implanted MTO. For ex vivo imaging, primary tumor, liver and lymph node were resected from tumor-bearing mice 2 min after intravenous injection of luciferin (5 mg/mouse; 88294, Thermo Scientific) and imaged using IVIS Lumina II (PerkinElmer). For in vivo imaging, mice were imaged using IVIS Lumina II thirty minutes after intraperitoneal injection of luciferin (5 mg/mouse; 88294, Thermo Scientific).

### BM and peripheral blood cells collection
Non-treated healthy mice or mice orthotopically implanted with MTO or MC38 were anaesthetized by isoflurane inhalation and their blood was collected from abdominal aorta. Collected blood was immediately mixed with 100 U heparin (FUJIFILM Wako). After the mice were euthanized, BM cells were collected from femur and humerus. Collected blood or BM were admixed with the Red Blood Cell Lysis Solution (130-094-183; Miltenyi Biotec). Total RNA was extracted from whole blood cells, BM cells or FACS-sorted cells (CD45⁺CD11b⁺CXCR4⁺, CD45⁺CD11b⁺CXCR4⁻, CD45⁺CD11b⁻, and CD45⁻ cells) using the RNeasy Mini Kit (QIAGEN), followed by qRT-PCR analyses (Supplementary Fig. 10e). The number of CD45⁺CD11b⁺ cells was counted by flow

cytometry by a FACSAria II (BD Biosciences). The following primary antibodies were used for flow cytometry: CD45 (1:70, 552848; BD Biosciences), CD11b (1:70, 553310; BD Biosciences) and CXCR4 (1:70, 146511; Biolegend).

### In vivo treatments
MTOs were orthotopically implanted in the rectum or injected to the spleen of 8–10 weeks old C57BL/6 and *Thbs1*⁻/⁻ mice. For CD8⁺ T cell depletion, mice were intraperitoneally injected with αCD8 antibody (200 μg; BE0061, BioXcell) or control IgG (200 μg; BE0090, BioXcell) twice per week from 5 weeks to 8 weeks after orthotopic implantation of MTOs in the rectum or from 0 weeks to 4weeks after splenic injection of MTOs. For CXCL12 inhibition, mice were intraperitoneally injected with LIT-927 (25 mg/kg; S8813, Selleck) three times per week for 8 weeks after orthotopic implantation of MTOs. For neutralizing antibody treatment, mice were treated with intraperitoneal injection of αPD-1 antibody (200 μg; BE0146, BioXcell), DC101 (αVEGFR2 antibody (800 μg; BE0060, BioXcell)) or control IgG (200 μg; BE0089, BioXcell) twice per week for 3 weeks starting at 5 weeks after implantation of MTOs. For FOLFOX treatment, mice were treated sequentially with intraperitoneal injection of oxaliplatin (6 mg/kg; HY-17371, MedChem Express), followed by 5-FU (50 mg/kg; F6627, Sigma-Aldrich) and levoleucovorin (90 mg/kg; HY-13667, MedChem Express) on a weekly basis for 3 weeks as previously described[50]. In the treatment experiments (αPD-1 antibody, αVEGFR2 antibody, and FOLFOX), the mice with less than 7 mm or more than 12 mm in tumor diameter at the beginning of drug treatment were defined as outliers and removed from the further analyses. The change in diameter of primary tumors from the beginning to the end of treatment was measured.

### In vitro stimulation of T cells
To check the mechanism of THBS1 to regulate T cell activity, splenocytes were collected from C57BL/6 or *Cd47*⁻/⁻ or *Cd36*⁻/⁻ mouse with the Tumor Dissociation Kit (130-096-730; Miltenyi Biotec) and were admixed with the Red Blood Cell Lysis Solution (130-094-183; Miltenyi Biotec). CD3⁺ T cells were sorted from the collected splenocytes by MACS using CD3ε MicroBead Kit (130-094-973; Miltenyi Biotec). Sorted CD3⁺ T cells were seeded at 10⁵ cells/well with the presence or absence of recombinant THBS1 (12.5 μg/ml, 25 μg/ml or 62.5 μg/ml for WT T cell, 25 μg/ml for *Cd47*⁻/⁻ or *Cd36*⁻/⁻ T cell; 7859-TH-050, R&D systems) simultaneously stimulated using anti-CD3/CD28 antibodies (11452D; Thermo Fisher) in 96 well plate. For analysis of IFNg activity, protein transport inhibitor (554724; BD Biosciences) was added when T cells were seeded in the plate. After 24 h of seeding, CD3⁺ T cells were collected and stained with following antibodies, fixed with Cytofix (554714; BD Biosciences), followed by flow cytometry by a FACSAria II (BD Biosciences). Active CD8⁺ T cells were selected as CD3⁺CD8⁺CD69⁺, CD3⁺CD8⁺IFNg⁺ or CD3⁺CD8⁺TCF7⁺ and inactive CD8⁺ T cell were selected as CD3⁺CD8⁺PDCD1⁺ or CD3⁺CD8⁺CTLA4⁺ (Supplementary Fig. 10b). The following primary antibodies were used for flow cytometry: CD3 (1:70, 555275; BD Biosciences), CD8 (1:70, 100706;

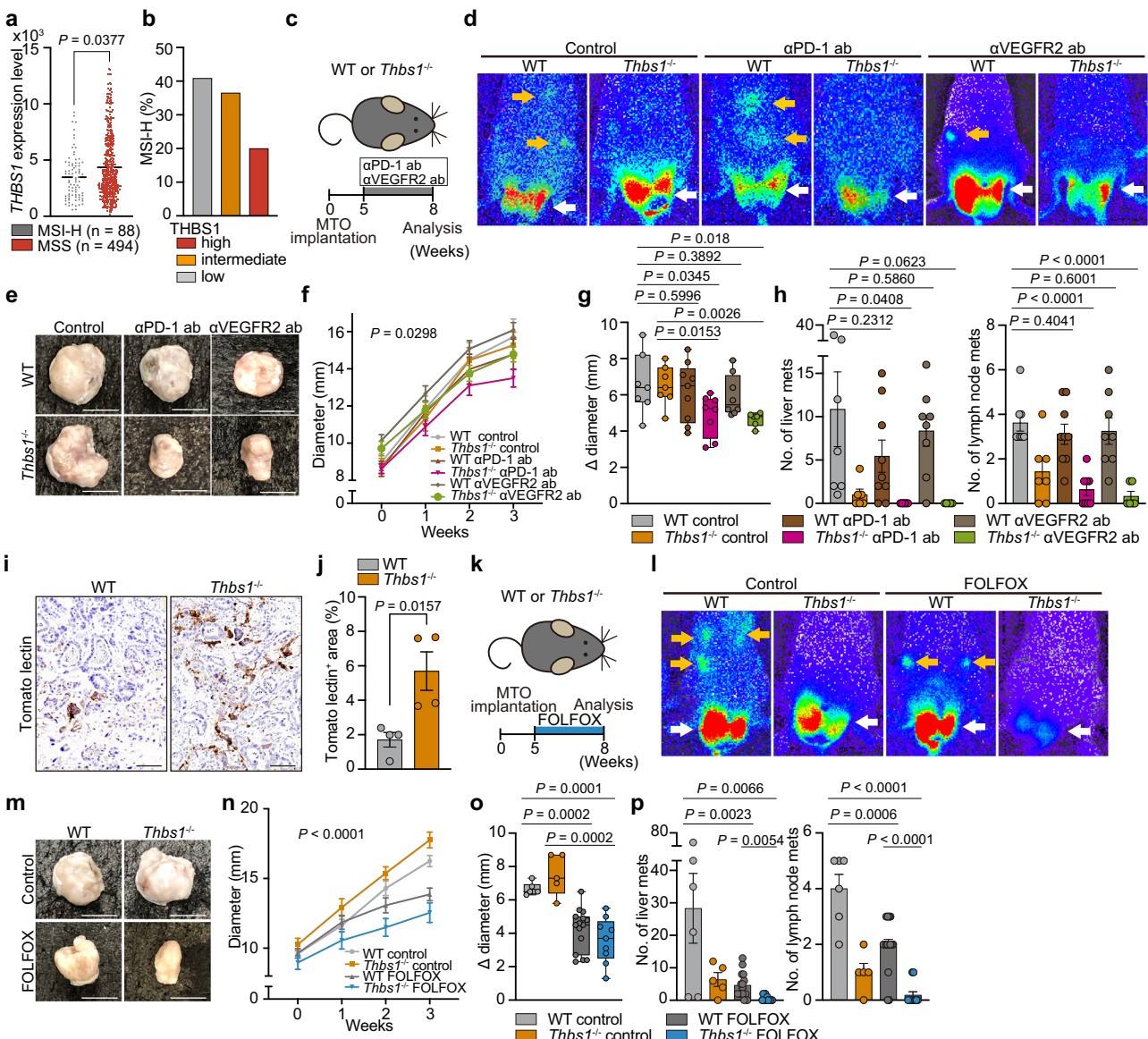

**Fig. 7 | Inhibition of THBS1 partially improved response of aggressive CRC to current treatments. a** *THBS1* expression in MSI-H or MSS CRC from TCGA. **b** Proportion of MSI-H CRC in TMA samples, stratified by THBS1 expression intensity (*n*: high = 100, intermediate = 115, low = 93). **c**–**h** Treatment experiment with anti-PD-1 antibody (αPD-1 ab) or anti-VEGFR2 antibody (αVEGFR2 ab) on MTO-bearing WT or *Thbs1*[-/-] mice (n: WT control = 8, *THBS1*[-/-] control = 7, WT with αPD-1 ab = 9, *Thbs1*[-/-] with αPD-1 ab = 8, WT with αVEGFR2 ab = 8, *Thbs1*[-/-] with αVEGFR2 ab = 6). Schematic representation **c**, and bioluminescence imaging **d**. Macroscopic images **e**, growth kinetics **f**, and the change in diameter **g** of primary tumors in indicated groups. Macroscopic numbers of liver and lymph node metastases **h**. **i, j** Tomato lectin staining (**i**) in the orthotopic MTO tumors in WT (*n* = 5) or *Thbs1*[-/-] mice (*n* = 4) and quantification **j**. **k**–**p**, Treatment experiment with FOLFOX on MTO-bearing WT or *Thbs1*[-/-] mice (*n* = mice per group: WT control = 6, *THBS1*[-/-] control = 5, WT with FOLFOX = 15, *Thbs1*[-/-] with FOLFOX = 9). Schematic representation **k** and bioluminescence imaging **l**. Macroscopic images **m**, growth kinetics **n**, and the change in diameter **o** of primary tumors in indicated groups. Macroscopic numbers of liver and lymph node metastases **p**. Arrows in **d, l**: primary lesions (white), distant metastases (yellow). Scale bars: 50 μm in **i**, 1 cm (rest). Mean ± SEM. The whiskers show smallest and largest values and the box extends from the 25th to 75th percentiles, and the line of the middle of the box is plotted at the median in Box and whiskers plot **g, o**. *P* values are calculated by two tailed, unpaired Student's *t* test **g, h, j, o, p**, two tailed, Mann–Whitney test **a**, or two-way ANOVA **f, n**. Source data are provided as a Source Data file.

Biolegend), CD69 (1:70, 104512; Biolegend), IFNg (1:70, 563376; BD Biosciences), TCF7 (1:70, 566692; BD Biosciences), PDCD1 (1:70, 135224; Biolegend) and CTLA4 (1:70, 106314; Biolegend).

**Cytotoxicity assay**

For cytotoxicity assay, OVA-expressing MTOs were generated by lentivirus infection (pLV[Exp]-Puro-EF1A > {Ovalbumin(ns)}: T2A:EGFP (VB900133-6569mtk); VectorBuilder). Effector T cells for OVA-expressing cells were obtained by stimulation of OVA53 cells by anti-CD3/CD28 antibodies (11452D; Thermo Fischer) overnight.

OVA-expressing MTOs were dissociated by Tripsin-EDTA (0.25%) (Thermo Fischer) and labeled with CFSE using 7-AAD/CFSE Cell-Mediated Cytotoxicity Assay Kit (600120; Cayman Chemical). $10^4$ cells of CFSE labeled OVA-expressing MTO and stimulated OVA53 cells ($10^2$, $5 \times 10^2$, $10^3$, $5 \times 10^3$, $10^4$ or $2 \times 10^4$ cells) were co-cultured in the presence or absence of recombinant THBS1 (25 μg/ml; 7859-TH-050, R&D systems) in 24 well plate for 20 h. After co-incubation, dead cells were stained with 7-AAD, followed by flow cytometry using a FACSAria II (BD Biosciences) (Supplementary Fig. 10c). Cytotoxicity was analyzed by the proportion of 7-AAD[+] cells in CFSE[+]

cells. The proportion was statically analyzed in each cell number condition of OVA cells.

## MDSC assay

For the examination of immunosuppressive function of MDSC, PMN-MDSC and MO-MDSC were collected from primary tumors of WT or *Thbs1*[−/−] mouse implanted MTO to the rectum using Myeloid-Derived Suppressor Cell isolation Kit (130-094-538; Miltenyi Biotec). Each MDSC ($10^4$ cells/well) and CD3[+] T cells ($10^4$ cells/well) collected from healthy WT mouse as described in (In vitro stimulation of T cells) were co-cultured with the stimulation by anti-CD3/CD28 antibodies (11452D; Thermo Fischer). After co-incubation, cells were collected and analyzed by flow cytometry using a FACSAria II (BD Biosciences) (Supplementary Fig. 10f). Immunosuppressive potency of each MDSC was analyzed by the proportion of CD69[+] cells or CTLA4[+] cells in CD8[+] cells. The following primary antibodies were used for flow cytometry: CD3 (1:70, 555275; BD Biosciences), CD8 (1:70, 100706; Biolegend), CD69 (1:70, 104512; Biolegend) and CTLA4 (1:70, 106314; Biolegend).

## Tube formation assay

To examine the potential function of CD36 in vascularization, tube formation assay was performed using Endothelial Tube Formation Assay (CBA-200; Cell Biolabs). 50 μl of ECM gel was added to each well of 96 well plate and incubated for 30 min. $5 \times 10^4$ cells of Mouse endothelial cells, C166 were harvested in each well in the presence or absence of recombinant THBS1 (25 μg/ml; 7859-TH-050, R&D systems), simultaneously treated with anti CD47 antibody (10 μg/ml; BE0270, BioXcell) or sulfosuccinimidyl oleate (SSO) (20 μM, 25 μM, 50 μM, 100 μM, 200 μM; 11211, Cayman Chemical). Images were obtained microscopically after incubation for 2 h. Cell images were obtained using microscope IX73 (Olympus). Quantification of tube area was performed by ImageJ/Fiji software v.2.3.0/1.53f[48].

## Generation of *Thbs1*-knockdown MTO

To examine the potential role of THBS1 expressed in tumor epithelium, *Thbs1*-knockdown MTOs were generated by induction of shRNA for *Thbs1* using lentivirus (sh*Thbs1*#1 (LVM(VB900138-8684hkq)-K1), sh*Thbs1*#2 (LVM(VB900120-5533jmq)-K1), scramble shRNA control (LVM(VB010000-0009mxc)-K1); VectorBuilder). The knockdown efficiency was confirmed by qRT-PCR for *Thbs1*. Proliferation potency of each organoid was analyzed by luciferase assay. For luciferase assay, $10^4$ cells of each organoid were seeded in matrigel in each well and cultured for 5 days and luminescence intensity was measured by SpectraMax iD5 (Molecular Devices) 10 min after 250 μg of luciferin (88294; Thermo Scientific) was added in each well every day. To examine potential role of THBS1 expressed in tumor epithelium for metastasis, splenic injection of *Thbs1*-knockdown MTO (sh*Thbs1*#2) or control MTO (Scramble shRNA) to WT mice was performed ($n = 6$).

## THBS1 ELISA

Human serum samples were collected from whole blood of patients with adenomas and CRCs by using EDTA tubes and centrifugation, and the concentration of THBS1 in the serum was analyzed by Human Thrombospondin-1 Quantikine ELISA Kit (DTSP10; R&D systems). Colorimetry was performed on a micro plate reader Sunrise (TECAN).

## Histology, immunohistochemistry, and immunofluorescence

Tissues were fixed 2–5 days in masked formalin2A (Japantanner) and embedded in paraffin. Paraffin-embedded tissues were cut into 5 μm sections. H&E staining was performed with standard protocols. Masson's trichrome staining was carried out following manufacturer's instructions (HT15-1KT; Sigma). For immunostaining, sections were deparaffinized, rehydrated, and then treated for antigen retrieval. After blocking in Protein Block Serum-Free solutions (DAKO), tissues were incubated with primary antibody overnight at 4 °C followed by

incubation with biotinylated (1:200, BA-5000, BA-1000, BA-2000, BA-4000; Vector Laboratories) or fluorophore-conjugated secondary antibody. For immunohistochemistry, endogenous peroxidase was quenched in 3% $H_2O_2$ in methanol for 10 min at room temperature. For immunohistochemistry, antibodies were visualized with avidin-biotin complex (Vectastain Elite; Vector Laboratories) using diaminobenzidine (DAB) (Dako) as the chromogen. For immunofluorescence analysis (double staining), sections were incubated with Alexa Flour 488 or 594 conjugated secondary antibodies (1:200, A21207, A21203, A21202, A21206; Thermo fisher) for 1 h at room temperature, followed by counterstaining with Hoechst (Thermo Fisher). For immunofluorescence staining, the signal was also visualized using the Opal 3-Plex Manual Kit (Akoya Biosciences), which allows simultaneous detection of multiple targets in the same image. Secondary antibodies (undiluted, ImmPRESS Reagent (MP-7401, MP-7404; Vector Laboratories), EnVision+ System-HRP Labelled Polymer (K4001; Dako)). Fluorophores Opal 520, Opal 570, and Opal 690 were used, and the sections were counterstained with Spectral DAPI. To investigate lectin perfusion, tumor sections were isolated from the mice orthotopically implanted MTO to the rectum 1 h after intravenous injection of biotinylated tomato lectin (100 μg/mouse, B-1175; Vector Laboratories) and were incubated with HRP-conjugated secondary antibodies (1:200, Vectastain Elite ABC HRP kits) for 1 h at room temperature and visualized with DAB (Dako). RNAscope was performed for detection of expression of *Thbs1* in MTO tumor sample using *Thbs1*-probes (No. 457891 (ACD)). The following primary antibodies were used for immunostaining: THBS1 (1:500, MA5-13398; Thermo Fisher), CD31 (1:100, 7769; Cell Signaling), CD8 (1:2000, ab209775; Abcam), CD4 (1:1000 (using Opal system), ab 183685; Abcam), CD11c (1:1000 (using Opal system), 97585; Cell Signaling), C-Cas3 (1:100, 9664; Cell Signaling), CK19 (1:500, ab52625; Abcam), CD11b (1:1000, ab133357; Abcam), GFP (1:500, ab6673; Abcam), NLRP3 (1:100, NBP2-12446; Novus Bio), αSMA (1:100, ab5694; Abcam), pSMAD3 (1:200, ab52903; Abcam), EPCAM (1:100, ab71916; Abcam), Ly6C (1:100 (using Opal system) 128002; Biolegend), Ly6G (1:1000 (using Opal system, 87048; Cell Signaling), F4/80 (1:2500 (using Opal system), 70076; Cell Signaling), FOXP3 (1:1000 (using Opal system), 12653; Cell Signaling), TCF7 (1:1500 (using Opal system), MA514965; Thermo Fisher), CD36 (1:1500 (using Opal system), ab202909; Abcam), CD47 (1:1000 (using Opal system), SC-53050; Santa Cruz), CXCR4 (1:4000 (using Opal system), ab181020; Abcam) and CTLA4 (1:1000 (using Opal system), ab237712; Abcam). For staining of EYFP, GFP antibody (1:500, ab6673; Abcam) was used. All images were collected from each stained slide on a BZ-X710 microscope (KEYENCE). The number of whole cells and positively stained cells in IHC images were measured using the QuPath[47], and the whole area and positively stained area in IF images were measured by ImageJ/Fiji software v.2.3.0/1.53 f[48].

## Flow cytometry analysis

Flow cytometry experiments were performed on BM cells, blood cells, tumor-consisting cells. BM cells and blood cells were collected from healthy mice or WT or *Thbs1*[−/−] mice orthotopically implanted MTO or MC38 in the rectum. Tumor-consisting cells were collected from primary tumors from WT or *Thbs1*[−/−] mice orthotopically inoculated MTO in the rectum or metastatic liver tumors from WT mice injected MTO to the spleen. For collecting tumor-consisting cells, fresh tumors were dissociated with Tumor Dissociation Kit (130-096-730; Miltenyi Biotec) and red blood cells were removed with using Red Blood Cell Lysis Solution (130-094-183; Miltenyi Biotec). The collected cells were blocked with mouse Fc Block (1:50, anti CD16/CD32 antibody; 553142, BD Biosciences) for 10 min on ice and then incubated with the following antibodies for 30 min on ice: CD45 (1:70, 552848; BD Biosciences), CD11b (1:70, 553310; BD Biosciences), CXCR4 (1:70, 146511; BioLegend), EPCAM (1:70, 48-5791-82; Invitrogen), CD45 (1:70, 557659; BD Biosciences) (1:70, 103106; Biolegend), CD3 (1:70, 555275; BD

Biosciences) (1:70, 100236; Biolegend), CD4 (1:70, 100406; Biolegend), CD8 (1:70, 553035; BD Biosciences) (1:70, 100706; Biolegend), CD69 (1:70, 104512; Biolegend), CD11c (1:70, 117318; Biolegend), CD11b (1:70, 553310; BD Biosciences), F4/80 (1:70, 565410; BD Biosciences), Ly6C (1:70, 560595; BD Biosciences), Ly6G (1:70, 562737; BD Biosciences), PDCD1 (1:70, 135224; Biolegend), CTLA4 (1:70, 106314; Biolegend) and CCR2 (1:70, 150610; Biolegend). For intracellular staining, fixation and permeabilization was performed using BD Cytofix/Cytoperm™ Fixation /Permeabilization Kit (554714; BD Biosciences) for staining of IFNg and Transcription Factor Buffer Set (562574; BD Biosciences) for staining of FOXP3 and TCF7, followed by incubation for 30 minuntes with the following antibodies: FOXP3 (1:70, 562996; BD Biosciences), IFNg (1:70, 563376; BD Biosciences), TCF7 (1:70, 566692; BD Biosciences) and Cy7/Alexa Flour 700-conjugated THBS1 (1:70). For generation of Cy7/Alexa Flour 700-conjugated anti THBS1 antibody, Cy7 Conjugation Kit (ab102882) or Alexa FlourR 700 Conjugation Kit (Fast)-Lightning-LinkR (ab269824; Abcam) was used with anti THBS1 antibody (MA5-13398; Thermo Fischer). Flow cytometry was performed on a FACSAria II (BD Biosciences) and the data was analyzed using FlowJo v.10. THBS1 positive cell sorting strategy is shown in Supplementary Fig. 10f. Each cellular population was determined as the following markers: CD8$^+$ T cell (CD3$^+$CD8$^+$), CD4$^+$ T cell (CD3$^+$CD4$^+$), Treg (CD3$^+$FOXP3$^+$), Dendritic cell (CD45$^+$CD11c$^+$), TAM (CD11b$^+$F4/80$^+$), monocyte (CD45$^+$CD11c$^-$CD11b$^+$Ly6C$^+$Ly6G$^-$F4/80$^-$), neutrophil (CD45$^+$CD11c$^-$CD11b$^+$Ly6C$^+$Ly6G$^+$F4/80$^-$), PMN-MDSC (CD45$^+$CD11c$^-$CD11b$^+$Ly6C$^+$Ly6G$^+$), MO-MDSC (CD45$^+$CD11c$^-$CD11b$^+$Ly6C$^+$Ly6G$^-$) (Supplementary Fig. 10a, d).

## Immunoblotting analysis
Tumors induced by orthotopic implantation were lysed in RIPA buffer (ab156034; Abcam) with phosphatase and protease inhibitors (5872; Cell Signaling) using gentleMACS Dissociator (Miltenyi Biotec). Protein concentration in lysates were determined by using Protein Assay Dye Reagent Concentration (5000006; Bio-Rad). Cell extracts were denatured, subjected to SDS-PAGE and transferred to membranes (Trans Blot Turbo, 1704156; Bio-Rad). After blocking with Blocking One (03953-95; Nacalai), the membranes were incubated with the following antibodies overnight at 4 °C: pSMAD3 (1:1000, 9520; Cell Signaling), SMAD3 (1:1000, 9513; Cell Signaling) and β-actin (1:4000, A1978; Sigma Aldrich). After 2 h incubation with the appropriate horseradish peroxidase-conjugated antibodies (1:10000, 7074/7076; Cell Signaling), the immune complexes were detected by chemiluminescence (34580; Thermo Fisher) on an Amersham Imager 600 (GE Healthcare).

## RNA extraction and qRT-PCR analysis
Total RNA from samples was extracted using the RNeasy Mini Kit (QIAGEN). Complementary DNA (cDNA) was obtained using the ReverTra Ace qRT-PCR RT Kit (TOYOBO). Real-time PCR was performed in duplicate using the SYBR Green PCR Master Mix (Roche Diagnostics) on a LightCycler 96 (Roche Diagnostics). Gene expression values for each sample were normalized to the 18 S RNA. qRT-PCR primers are listed in Supplementary Table 5.

## RNA-Seq and analysis
For RNA-seq, total RNA was extracted from the tumors generated by orthotopic implantation of MTO in the rectum of WT or *THBS1$^{-/-}$* mice using the RNeasy Mini Kit (QIAGEN). RNA-seq was performed by Macrogen, Inc on the Novaseq6000 platform with 2 × 100 base pair paired-end sequencing using SMART-Seq v4 Ultra Low Input RNA kit and TruSeq RNA Sample Prep Kit v2. Adaptors and low-quality bases were trimmed from the reads using Trimmomatic (version 0.39) with default parameters. Reads were mapped to the *Mus musculus* reference genome build mm10 using STAR (version 2.7.3a) and counted by RSEM (version v1.3.1). Read count data were normalized using the iDEGES/edgeR method. Normalized count data were used for gene set enrichment analysis (GSEA) and differentially expressed genes were determined using TCC (Tag Count Comparison) with a false discovery rate cutoff value < 0.05. GSEA was performed using GSEA software (http://www.gsea-msigdb.org/gsea/index.jsp).

## Single-cell RNA sequencing
Tumors generated by orthotopic implantation of MTO in the rectum of WT and *Thbs1$^{-/-}$* mice ($n$ = 2 mice per group) were processed into single-cell suspension using Tumor Dissociation Kit (Miltenyi Biotec). Dead cells were removed by Dead Cell Removal Kit (Miltenyi Biotec). scRNA-seq libraries were generated using the Chromium Single Cell 3' v3 Reagent (10× Genomics). Cells were loaded onto the 10× Chromium Controller (10× Genomics) using Chromium Next GEM Chip G single cell kit (10× Genomics) at a concentration of 1,000 cells per μl as described in the manufacturer's protocol (10× Chromium Single Cell 3' Reagent Kits User Guide (v3.1 Chemistry Dual Index)). 5,000 cells from each WT mouse (W1, W2: $n$ = 2) and 10,000 cells from each *Thbs1$^{-/-}$* mouse (T1, T2: $n$ = 2) were loaded. Generation of gel beads in emulsion (GEMs), barcoding, GEM-RT clean-up, complementary DNA amplification and library construction were all performed as per the manufacturer's protocol. Individual sample quality was checked using a TapeStation (Agilent Technology). The final library pool was sequenced by a DNBSEQ-G400RS (MGI). The Cell Ranger version 6.1.1 software suite from 10× Genomics was used to process, align and summarize unique molecular identifier (UMI) counts. Estimated number of cells of T1, T2, W1 and W2 were 9875, 6791, 5071, and 5268 respectively. Mean depth of T1, T2, W1 and W2 were 31720, 52408, 67364, and 58361 reads per cell and 800, 1088, 1308, and 1446 median genes per cell, respectively. Additional data processing and analysis was performed using Seurat version 4.2.3[51] in R version 4.2.3. Cells expressing fewer than 200 genes and cells expressing greater than 20% mitochondrial related genes were removed. The genes expressed in fewer than five cells were excluded. The expression set of selected 8209 cells from T1, 4907 cells from T2, 3852 cells from W1, 3963 cells from W2 was converted into Seurat object. Each sample was normalized using SCTransform[52]. For integration, the Seurat's SelectIntegrationFeatures function was used in the merged object to select genes taken as input in anchors identification procedure by the Seurat's PrepSCTIntegration function and the Seurat's FindIntegrationAnchors function. The Seurat's IntegrateData function was used to integrate all objects. Dimensionality reduction was performed by RunPCA function and RunUMAP function. All cells were classified into three compartments, epithelium (*Epcam*), Immune (*Ptprc*), or stroma (*Pecam1*, *Dcn* or *Rgs5*). Immune compartment was extracted and re-normalized. Then it was devided into B cells (*Cd79a*), T/NK cells (*Cd3d* or *Nkg7*) or Myeloid cells (*Lyz2*, *Itgam* or *S100a8*). *Cd8a* positive cluster was extracted from T/NK and proceeded to further analysis. Representative genes for clusters of myeloid cells and CD8$^+$ T cells are shown in Supplementary Fig. 3d, g, respectively. Representative genes for stem-like signature, dysfunction signature and M2-macrophage-related signature are shown in Supplementary Table 6. Information of recipient mouse genotype in published single cell transcriptome data of orthotopic MTO tumor (GSE154863) was provided by the original author (A.D.). Cells of WT mice were used for analysis. Cells expressing greater than 7% mitochondrial related genes were removed. The myeloid cell cluster was detected as described above and in Fig. 5j.

## Bioinformatics analysis
CRC samples in TCGA were stratified according to CMS subtypes ($n$: CMS1 = 85, CMS2 = 132, CMS3 = 78, CMS4 = 184) based on previous report using CMScaller package[5,53]. MSI-H and MSS groups in TCGA were stratified by MSIsensor ($n$: MSI-H = 88, MSS = 494)[54]. Bulk iCMS classification was performed according to previous report[16] using R4.2.3 and CMScaller package[53]. Briefly, gene expression quantities

standardized with FPKM was log transformed and iCMS was determined using nearest template prediction. Adjusted *p*-value < 0.05 was considered as significant. IMF classification was conducted with applying iCMS, MSI, and fibrotic transcriptome (CMS4), and CRC samples in TCGA were stratified into five IMF classes: i2_MSS_NF ($n = 117$), i2_MSS_F ($n = 101$), i3_MSI ($n = 81$), i3_MSS_NF ($n = 73$) and i3_MSS_F ($n = 51$). The "CMS4_UP" and "CMS4_DOWN" signatures were generated, as previously described[4,17]. Representative genes for "CMS4_UP" and "CMS4_DOWN" signatures are shown in Supplementary Table 6. For analysis of CXCL12-expressing cells in single cell transcriptomic dataset of human CRCs (SMC), we extracted B cell (CD79A), epithelial cell (EPCAM), mast cell (KIT), myeloid (CD68), T cell (CD3D) and stromal cells (DCN) compartment in accordance with original author's annotation[30] and re-clustered stromal cells into five subpopulations: fibroblast (COL1A2, PDPN, DCN), smooth muscle cell (MYH11), pericyte (MCAM, RGS5), endothelial cell (PECAM1, LYVE1) and enteric glial cell (S100B). For analysis of myeloid cells, myeloid cell compartment was extracted in accordance with original author's annotation[30]. To correct batch effect, myeloid cell object was split by sample ID with Seurat's SplitObject function and normalized with SCTransform method. Then data was integrated as described above. Dimensionality reduction was performed by RunPCA function and RunUMAP function. Re-integrated myeloid cells were stratified into six clusters according to expression of known marker genes. The marker genes used here are shown in Fig. 5m.

### Statistics and reproducibility

Pairwise differences were measured using two-tailed independent student's t-tests. If the data did not meet this test, a Mann-Whitney U-test was used. Statistical significance between groups of 3 or more was determined by a one-way or two-way ANOVA, followed by the Tukey's multiple comparison test. Data are presented as the mean ± SEM. Survival was measured by the Kaplan-Meier method and analyzed by Log-rank (Mantel-Cox) test. Outlier was identified and removed using the ROUT method (Q = 1%). Statical correlation was measured using the Pearson Correlation Coefficient (two-tailed, confidence interval (CI) = 95%). Logistic regression analysis was employed to estimate univariate and multivariate odds ratio and 95% CI. Statistical analysis was performed using GraphPad Prism 9 or JMP. Values of $P < 0.05$ were considered as significantly different.

### Reporting summary

Further information on research design is available in the Nature Portfolio Reporting Summary linked to this article.

## Data availability

The data generated in this study are available in the Genomic Expression Archive (GEA) database under the accession numbers E-GEAD-561 for bulk RNA-seq [https://ddbj.nig.ac.jp/public/ddbj_database/gea/experiment/E-GEAD-000/E-GEAD-561/] and E-GEAD-562 for scRNA-seq data [https://ddbj.nig.ac.jp/public/ddbj_database/gea/experiment/E-GEAD-000/E-GEAD-562/]. Data for TCGA-COREAD was accessed through cBioportal (https://www.cbioportal.org). Raw gene expression data of CRC patient dataset (GSE35602, GSE14333 and GSE17536) were directly accessed through the GEO website (NCBI). Raw gene expression (count) data and metadata of single cell transcriptomic dataset of human CRCs (SMC) were directly accessed through the GEO website (GSE132465), and converted into Seurat object. Published single cell transcriptome data of orthotopic MTO tumor was downloaded from GEO website (GSE154863), and converted into Seurat object. The remaining data are available in the Article, Supplementary Information or Source Data file. Source data are provided with this paper.

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

## Acknowledgements

Research was supported by the Project for Cancer Research and Therapeutic Evolution (P-CREATE, 18cm0106142h0001 (A.F.), 20cm0106177h0001 (H.S.), 20cm0106375h0001 (A.F.), 21cm0106283h0001 (Y.N.)), P-PROMOTE (22cm0106283h0002 (Y.N.), 22ama221515h0001 (H.S.)), and PRIME (20gm6010022h0003 (A.F.)) from the Japan Agency for Medical Research and Development (AMED); the Fusion Oriented Research for disruptive Science and Technology (FOREST, 23719768 (Y.N.)); Moonshot Research and Development Program (JPMJMS2022-1), and COI-NEXT (JPMJPF2018); the JSPS KAKENHI (JP19H03639 (A.F.), JP20K22841 (Y.H.), JP20H03659 (H.S.), JP21K19480 (H.S.), and JP21H02902 (Y.N.)); the foundations of Takeda Science (Y.N.), Princess Takamatsu Cancer Research (Y.N.), Astellas, Daiichi Sankyo of Life Science (Y.N.); JST SPRING (JPMJSP2110 (M.O.)). We thank Mason Freeman (Harvard Medical School) for providing *CD36⁻/⁻* mice and Toshiko Sato for technical assistance. A part of this study was conducted through the CORE Program of the Radiation Biology Center, Kyoto University.

## Author contributions

M.O. and Y.N. coordinated the project and designed the experiments with help from K.Mi., M.K., G.Y., Mi.N., K.H., Y.F., M.S., T.Mas., O.A., Mu.N., T.Y., Sa.O., Y.H., M.T. and A.F. M.O., K.I. and T.Mar. performed the experiments with help from A.H., T.K. and K.Kab. N.Ao. analyzed TMA data with help from Q.H. and A.M-O. N.Ag., T.U., H.K., K.Kaw., M.Y., K.Ma., Y.S., T.Mat., K.O., and N.S. provided materials that made the study possible. M.O. and K.I. performed scRNA-seq with help from N.K., M.M.N. and Se.O. M.O., K.I. and Y.N. analyzed the data with help from Y.M. and A.D. M.O and Y.N. wrote the manuscript with help from A.F., M.T-M., J.M. and H.S.

## Competing interests

Yuichi Fukunaga is employed by Sumitomo Pharma Co., Ltd. The remaining authors disclose no conflicts.

## Additional information

**Mayuki Omatsu**[1], **Yuki Nakanishi** [1] ✉, **Kosuke Iwane**[1], **Naoki Aoyama**[1], **Angeles Duran**[2], **Yu Muta** [2,12], **Anxo Martinez-Ordoñez**[2], **Qixiu Han**[2], **Nobukazu Agatsuma**[1], **Kenta Mizukoshi**[1], **Munenori Kawai**[1], **Go Yamakawa**[1], **Mio Namikawa**[1], **Kensuke Hamada**[1], **Yuichi Fukunaga**[1,3], **Takahiro Utsumi**[1], **Makoto Sono**[1], **Tomonori Masuda** [1], **Akitaka Hata**[4], **Osamu Araki**[1], **Munemasa Nagao** [1], **Takaaki Yoshikawa**[1], **Satoshi Ogawa** [1], **Yukiko Hiramatsu** [1], **Motoyuki Tsuda**[1], **Takahisa Maruno**[1], **Toshiaki Kogame**[4], **Hiroaki Kasashima** [5], **Nobuyuki Kakiuchi** [1,6], **Masahiro M. Nakagawa** [7], **Kenji Kawada** [8], **Masakazu Yashiro**[5], **Kiyoshi Maeda**[5], **Yasuyuki Saito** [9], **Takashi Matozaki** [9,10], **Akihisa Fukuda**[1], **Kenji Kabashima** [4], **Kazutaka Obama**[8], **Seishi Ogawa**[7], **Nader Sheibani** [11], **Maria T. Diaz-Meco** [2], **Jorge Moscat** [2] & **Hiroshi Seno**[1]

[1]Department of Gastroenterology and Hepatology, Kyoto University Graduate School of Medicine, Kyoto, Japan. [2]Department of Pathology and Laboratory Medicine, Weill Cornell Medicine, New York, USA. [3]Cancer Research Unit, Sumitomo Pharma Co., Ltd, Osaka, Japan. [4]Department of Dermatology, Kyoto University Graduate School of Medicine, Kyoto, Japan. [5]Department of Gastroenterological Surgery, Osaka Metropolitan University, Osaka, Japan. [6]The Hakubi Center for Advanced Research, Kyoto University, Kyoto, Japan. [7]Department of Pathology and Tumor Biology, Kyoto University, Kyoto, Japan. [8]Department of Gastrointestinal Surgery, Kyoto University, Graduate School of Medicine, Kyoto, Japan. [9]Division of Molecular and Cellular Signaling, Department of Biochemistry and Molecular Biology, Kobe University Graduate School of Medicine, Kobe, Japan. [10]Division of Biosignal Regulation, Department of Biochemistry and Molecular Biology, Kobe University Graduate School of Medicine, Kobe, Japan. [11]Department of Ophthalmology and Visual Sciences, University of Wisconsin-, Madison, Wisconsin, USA. [12]Present address: Department of Gastroenterology and Hepatology, Kyoto University Graduate School of Medicine, Kyoto, Japan. ✉e-mail: yuki@kuhp.kyoto-u.ac.jp

