## [Peer Review File · Nature Communications]

THBS1-producing tumor-infiltrating monocyte-like cells contribute to immunosuppression and metastasis in colorectal cancerREVIEWER COMMENTS

Reviewer #1 (Remarks to the Author): with expertise in colorectal cancer

In this manuscript “Tumor-infiltrating monocyte-like cells create immunosuppressive and therapy-resistant microenvironment via THBS1 production in colorectal cancer”, the authors present a model in which tumor CXCL12/CCR4 stimulates bone marrow monocytes to express THBS1 that is then able to bind CD47/CD36 cells in the tumor bed and create an immunosuppressive TME. At least, that is how I interpret their data. They use mouse tumor organoid models (MTO), generated from Apc, Trp53, Kras, Tgfbr2 GEM, and MC38 mice generated from carcinogen-induced colon cancer in C57BL/6. Tumors are injected into the rectum of syngeneic mice and straight THBS1 knock-out mice in which THBS1 is eliminated in all cells.

This reviewer would like clarification of some of the experimental details.

What cells in the TME express THBS1? RNAscope would answer this question. If THBS1 is expressed in the tumors themselves, this would complicate using the THBS1 knock-out mice. Might need to show by IHC that THBS1 is not present in tumors. A query of the protein atlas can give validity to the claim that monocytes are the main actors for THBS1 effects.

Along with CD14, it would be helpful to have other myeloid lineage markers – for example, CD68, CD11B. The authors should mention that CMS4 is associated with an inflammatory signature and stromal activation but show that THBS1^{-/-} mice have GSEA data with increased enrichment in inflammatory or immune response - might need to explain this and how these aren't in opposition for the reader.

THBS1 has pleiotropic effects. The authors seem focused on the immunosuppressive effects. The authors might consider performing co-culture experiments with a T cell killing assay with tumor cells incubated with or without THBS1.

Can the authors confirm that CD47 and CD36 are only receptors for THBS1? And what cells express these receptors? This is necessary for their claims that CD47 and CD36 knock-out

mice phenocopy the THBS1 knock-out mice. Showing a difference in activation of downstream signaling based on addition or absence of the ligand might be helpful, as well as colocalization studies/in vitro results looking at the markers as well as vascularization assays might enhance this mechanistic insight.

The authors mention THBS1 was increased in CMS4 and CCS3 CRC subtypes. Colorectal Cancer Subtypes (CCS) classification is not widely used. Can the authors state the value of using this classification system in this report? They should update their CMS classification (I Jonito et al., Nature Genetics 54:963-975, 2022).

The authors are encouraged to examine whether THBS1 is increased in MSS vs MSI-H CRC.

Reviewer #2 (Remarks to the Author): with expertise in colorectal cancer, metastasis

This manuscript by Omatsu et al. investigated the promoting role of THBS1 in the colorectal cancer by forming the immunosuppressive and vascular-rich tumor environment. The authors first identified that high THBS-1 expression in tumor stroma was correlated with poor survival in human CRC patients. Then they verified those observations in mouse CRC models. By using Thbs1 knock out mice, they found that loss of TME-derived THBS1 suppressed CRC liver and lymph node metastasis. Next, they performed ScRNA-seq analysis from the MTO tumors from WT and Thbs1 knock out mice, revealed that Thbs1 deficiency can block T cell exhaustion and increase the stem cell-like CD8+ T cells. By using Cd36 knock out and Thbs1 knock out mice, they concluded that THBS1-CD47 and THBS1-CD36 axes were both responsible for the formation of immunosuppressive microenvironment, while THBS1-CD36 can further influence the vascular structure. Finally, they identified the primary source of THBS1, and demonstrated that THBS1-expressing myeloid cells were recruited from bone marrow to TME via CXCL12-CXCR4.

The authors are to be recommended on a very extensive array of experimentation, the use of different transgene mice and the numerous innovative approaches. However, the data for supporting the contribution of myeloid cell-derived THBS1 in CRC are limited, and there are some confusing points that either explanation or further experimentation.

1. The authors found that THBS1 expression is mainly in the stroma of CRC samples by IHC staining in Fig.1a, while the mRNA expression of THBS1 were analyzed in Fig.1d by using TCGA datasets, in which were mainly tumor tissues. Therefore, the bioinformatic analysis in TCGA is not representative, also for the survival analysis. It is better to use your own TMA array to do the survival analysis.

2. Fig.1i, the authors revealed the relationship between THBS1 expression and masson/CD14 by showing IHC staining. For the same CRC samples, it is better to do the staining on the consecutive sections, which will be more representative.

3. The study focus on the TME-derived THBS1 in CRC. The CRC cells can also produce THBS1, the role of tumor cell-derived THBS1 should be investigated and excluded. How is the THBS1 expression in tumor cells? The expression of THBS1 in organoid and MC38 cells should be detected. Whether the tumor cell-derived THBS1 can contribute CRC progression?

4. For the bioluminescence imaging, such as. Fig.2e, 5e,6i, 6r, 7d, 7i, 7q, the authors showed the primary lesion(white), lymph node (orange), and liver (yellow) metastasis. How did the authors define the signaling are in the lymph node or liver? It is better to combine CT with IVIS imaging to better define the lymph node and liver. Or to isolate the lymph node and liver, to do the ex -vivo imaging, to check the luciferase signaling from lymph node and liver.

5. Fig.3j, the authors performed the stimulation experiment on CD3+T cells with and without recombinant THBS1 treatment (25ug/ml). How did the authors define the concentration of recombinant THBS1? Is this concentration-dependent effect? Why using the CD3+T cells? It is better to isolate the CD8+T cells to perform this experiment. For the activation of T cells, normally both anti-CD3 and anti-CD28 were used. Why did the authors use anti-CD3 only, which is not enough to activate T cells? In addition, the protein expression of IFN-g and CTLA4 should be checked by flow cytometry, which makes more sense.

6. Fig.6f, the magnified region should be marked in the figure, to make it more visible and

obvious.

7. Sup Fig.4c, the full WB blot should be provided, and the molecular weight should be marked.

8. Sup Fig5a, Sup Fig.6h, the scale bars for THBS1 and CD31 staining were different.

Reviewer #3 (Remarks to the Author): with expertise in bioinformatics, immunology

Omatsu, M, et al, present a study of the immunosuppressive mechanisms in an aggressive subtype of colo-rectal cancer (CRC), integrating mouse models, scRNAseq of a limited set of human patient samples, and bulk RNAseq and IHC of large human patient cohorts. In particular, they demonstrate a central role for thrombospondin-1 (THBS-1) produced by monocytes recently recruited from the bone marrow to the tumor site. THBS-1 expression is shown to be associated with poor prognosis in human tumors. Mouse experiments show a clear and strong effect on vascularization, T cell infiltration, necrosis, and metastasis and a potential, although not striking, effect on efficacy of immune checkpoint blockade, chemotherapy, and anti-angiogenesis therapy.

In summary, this manuscript reveals a clear mechanism in a frequent, aggressive cancer for recruitment of bone marrow derived myeloid populations, their contribution to generating an immunosuppressive microenvironment, and their relationship with metastasis. This could be appropriate for Nature Communications however there are a few major issues (see below) that need to be addressed before this is clear.

Major comments

1. The correlative analyses in Figure 1 and Supplementary Figure 1 connecting THBS1 expression with eg prognosis, CRC subtype, gene expression profiles, etc need to more clearly demonstrate how much THBS1 independently drives these patterns versus just being a marker of the CMS4 subtype.

Additionally, much more justification/information is needed for the splitting of samples by

tiers of THBS1 expression. In Fig 1 d-f, why is it not just by median ? In Fig 1 o, why are intermediate and high grouped together ?

2. The timing in Figure 2 is not clear. If tumor is implanted at week 8, what does week 0 refer to ? This is particularly important for interpreting the survival analysis in panel h. Given the strong effects of Thbs1-/- early in Figure 2, especially e-g, the KM curves in panel h are very surprising. Can the authors explain why there is just a modest delay and then all the Thbs1-/- die as rapidly as the WT ? Are they dying of the disease in a different way ?

3. In all the scRNAseq analyses, more extensive explanation/justification must be provided for all cluster annotations (eg with heatmaps, differential gene expression analysis, marker genes, or other approaches).

4. The results in Fig 7 on THBS1 inhibition in combination with other treatments do not demonstrate a compelling synergistic effect and therefore should not be highlighted in the title and abstract. Relatedly, survival analysis and tumor growth kinetics are standard analyses in these types of in vivo tumor treatment models – why are they not included here ? The evidence connecting THBS1 to metastatic activity (eg Fig 2g, 4e, 5f, 6h and to a lesser extent 6s), conversely, is much more compelling and perhaps does warrant highlighting early and clearly.

Minor comments

1. (line 108) “These results suggest that the stroma-infiltrating cells are the primary source of THBS1 and demonstrate a strong association between high THBS1 level, aggressive phenotype, and poor prognosis in human CRCs” and Fig. 1c

The expression of THBS1 appears higher in the normal colon than in the tumor epithelium. Is it significantly different? Can you provide a possible explanation? Additionally, the violin plots should be replaced by individual data points since n is relatively low. Finally, at this point in the manuscript it is demonstrated that THBS1 expression comes from the stroma, but not that it is from infiltrating cells.

2. Fig. S 1d, e

Why are there different n for the THBS1 high and low categories between these two plots from the same dataset/cohort?

3. Fig. 2h,i,j

Can these be re-sized to avoid having the awkward “x10” above the vertical axes in the plot at far right? It is currently confusing to compare with results of similar experiment in panel c.

4. Fig. 3c

Although the statistical test (Mann-Whitney, aka Wilcoxon) indicates the differential expression is significant, the violin plots are not convincing. Perhaps another type of visualization would be more useful? Ideally, validation by FACS would be even better.

5. (line 197) and Fig. 3k

Please provide a citation for these genes as IFN γ target/marker genes. Similar for all signatures/gene sets analyzed in the paper.

6. (line 231-232) and Figure 4

The phenocopies between *Thbs1*^{-/-}, *CD36*^{-/-}, and *CD47*^{-/-} are clear. However, attributing this directly to interactions between THBS1 and these proteins and referring to “THBS1-CD47 and THBS1-CD36 axes” seems premature without double mutants since, at least a priori, other receptors for THBS1 or other ligands for CD47 and CD36 could explain these effects through parallel rather than direct mechanisms.

7. Fig. 5i-k

These same genes should also be shown in the mouse scRNAseq data.

8. Fig. S 5b

As with major comment 1, provide these plots divided by CRC categories.

9. Fig. 6b

Please provide the total number of cells in each condition in addition to the percentage to rule out eg a decrease in lymphocytes.

10. (lines 303-304) Fig. 6l and Supplementary Figure 6g

Please provide statistical tests that these genes are specifically upregulated in mesenchymal CMS4 CRCs.

11. (lines 303, 307) Fig. 6m and Supplementary Figure 6g

It is confusing that CCL2 is high in CMS4 from TCGA but then lower in the MTO compared to MC38. Can the authors provide any hypotheses?

12. Fig. 7

Why does the data for the WT and Thbs1^{-/-} control conditions appear identical in panels f and s, but different from same conditions in panel n? All violin plots in this figure should be replaced by individual data points. The evidence that combination therapy is synergistic or even additive here, outside of potentially panels e and r, is not compelling.

Reviewer #4 (Remarks to the Author): with expertise in cancer immunology, myeloid cells

Omatsu et al report that a high expression of thrombospondin-1 (THBS1) positively correlates with the mesenchymal characteristics, immunosuppression, and poor prognosis in CRCs. They found that bone marrow derived monocyte-like cells recruited by the action of CXCL12 are the primary source of THBS1

contributing to the creation of a therapy-refractory environment by inducing cytotoxic T cell exhaustion and impairing vascularization. According to the authors, THBS1 loss in the TME renders tumors sensitive to immune checkpoint inhibitors and anti-cancer drugs. However, there are several critical points that need to be addressed to consider this study suitable for publication in Nature Communications:

1. The authors observed a substantial increase in CD8⁺ T cell infiltration and cleaved caspase-3 (C-Cas3)⁺ apoptotic cells in orthotopic tumors from Thbs1^{-/-} mice. Treatment with anti-CD8 antibody effectively depleted CD8⁺ T cells and, concomitantly, decreased

apoptotic cells in the primary tumors (Fig. 2i,j), confirming the cytotoxic activity of the infiltrating CD8⁺ T cells. How do the authors explain that there is no significant difference in tumor growth in KO mice? If THBS1 contributes to immunosuppression, why are T cells not able to kill tumors in KO mice?

2. Thbs1^{-/-} mice showed increased vascular formation, thus I was wondering if there is a change in the myeloid compartment that is known to contribute to the immunosuppressive TME. No changes in immunosuppressive myeloid cells infiltration may explain why T cells are still ineffective in killing tumors in those KO mice.

3. The authors showed reduced metastatic lesions, therefore it would be critical to compare the immune TME and THBS1 levels in primary tumor vs metastatic lesions. Such differences in immune composition and THBS1 levels may explain why there is reduction in metastasis lesions only.

4. The authors posited that increased infiltrating CD8⁺ T cells in the orthotopic primary site may account for the reduction of metastatic burden in MTO-bearing Thbs1^{-/-} mice. It is also possible that there is an increase of CD8 T cells in metastatic sites that may help to reduce metastatic burden. Thus, what is the frequency of T cells in metastatic lesions in those KO mice? The authors showed that loss of THBS1 blocks T cell exhaustion and increases stem-like CD8⁺ T cells, what is the phenotype of T cells in primary vs metastasis?

5. BM-derived monocyte-like cells are the primary producers of THBS1 in stroma-rich intestinal tumors. THBS1 affects the activation of T cells in vitro, therefore monocyte like cells may suppress T cell activation via the release of THBS1. Since myeloid cells used different mechanisms to suppress T cells, I was wondering if myeloid cell derived THBS1 contributes to the immunosuppressive activity of these cells. Moreover, what is the expression of other suppressive markers in WT vs KO mice?

6. The authors found the highest Thbs1 expression in CD45⁺CD11b⁺CXCR4⁺ cells in BM of MTO-bearing mice, suggesting that THBS1-expressing CD11b⁺ cells are recruited via the action of CXCL12-CXCR4 signaling. The authors suggest that these cells

(CD45+CD11b+CXCR4+) are monocyte like cells. However other myeloid cells expressed these markers. Therefore to better understand the myeloid cell composition, the authors should characterize CD45+CD11b+CXCR4+ cells in primary tumors and metastasis sites.

7. CD8+ T cells in 176 Thbs1^{-/-} tumors exhibited a decrease in dysfunctional markers such as Pcd1 and Ctla4. How do the authors explain that anti-PD1 therapy may work in this model? Do the mice survive longer? Thbs1^{-/-} tumors and combo with anti-PD1 showed similar reduction in mets. However, tumors are smaller in KO mice following treatment with anti-PD1. How do the authors explain that a reduction in primary tumor does not reflect in a significant decrease in number of mets? I have observed a similar trend in fig7m and s.

loss of THBS1 activated the antitumor immune response and suppressed metastasis (new Fig. 5b–g and Supplementary Fig. 7a, 7b).

2. Along with CD14, it would be helpful to have other myeloid lineage markers – for example, CD68, CD11B.

Following this reviewer's comment, we have performed IHC for other myeloid lineage makers CD68 and CD11B in TMA of human CRC and found clear correlative relationship between expression of THBS1 and that of CD68 and CD11b (new Supplementary Fig. 1j).

The authors should mention that CMS4 is associated with an inflammatory signature and stromal activation but show that THBS1^{-/-} mice have GSEA data with increased enrichment in inflammatory or immune response - might need to explain this and how these aren't in opposition for the reader.

Thank you for pointing this out. According to the original paper on consensus molecular subtyping, CMS1 CRC is characterized by an active immune response with high infiltration of effector immune cells, whereas CMS4 CRC is characterized by increased stromal activation and immune infiltration (Guinney J et al., *Nat Med*, 2015, Fig. 3i). This suggests that the TME in both CMS1 and CMS4 show immune cell infiltration and enrichment of inflammatory signatures, distinct from that of CMS2 and CMS3, which display few or no infiltrating immune cells. However, in contrast to the highly immunogenic TME of CMS1 CRC, which contains a large number of effector immune cells such as cytotoxic CD8⁺ T cells, the inflamed TME of CMS4 CRC includes a substantial number of immunosuppressive cells such as MDSCs, TAMs, and regulatory T cells (Guinney et al., *Nat Med*, 2015; Dienstmann et al., *Nat Rev Cancer*, 2017), indicating that the types of inflammation in CMS1 (immune active) and CMS4 (immunosuppressed) are completely different. Orthotopic tumors in *Thbs1*^{-/-} mice exhibited increased infiltration of CD8⁺ T cells (new Fig. 2f,3m and Supplementary Fig. 2n,5a) and increased enrichment of gene signatures, including the IFN gamma response (GSEA data, Fig. 2c and Supplementary Fig. 3i), suggesting that THBS1 inhibition switches the state of inflammation from “immunosuppression” to “immune active.” In the response to this comment, we have explained in more detail the characteristics of inflammation in the TME of mesenchymal CMS4 in revised manuscript and avoided confusing descriptions that could have misled the reader.

3. THBS1 has pleiotropic effects. The authors seem focused on the immunosuppressive effects. The authors might consider performing co-culture experiments with a T cell killing assay with tumor cells incubated with or without THBS1.

As suggested by the reviewer, we performed a T cell-killing assay using T cells and OVA-expressing MTO. THBS1 significantly attenuated the cytotoxic activity of CD8⁺ T cells to tumor cells (Fig. 2q).

4. Can the authors confirm that CD47 and CD36 are only receptors for THBS1? And what cells express these receptors? This is necessary for their claims that CD47 and CD36 knock-out mice phenocopy the THBS1 knock-out mice. Showing a difference in activation of downstream signaling based on addition or absence of the ligand might be helpful, as well as colocalization

studies/in vitro results looking at the markers as well as vascularization assays might enhance this mechanistic insight.

Under physiological conditions, THBS1 interacts with multiple receptors, including CD47, CD36, and TGFbeta; however, in the context of mesenchymal CRC, we propose that THBS1 exerts its immunosuppressive function by binding to CD47 and CD36 on CD8⁺ T cells. Regarding the cell types expressing CD47 and CD36, scRNA-seq of human CRC demonstrated that CD47 expression was almost ubiquitous in a variety of immune, epithelial, and stromal cells, whereas that of CD36 was observed in immune and stromal cells (see figure below). Among immune cells, CD36 is primarily expressed by myeloid cells, and also by some CD8⁺ T cells. Double immunofluorescence staining revealed that infiltrating CD8⁺ T cells in MTO tumors expressed both CD47 and CD36 (new Fig. 4g).

Response to Reviewers Fig. 2 scRNA-seq data of human CRC (top) showed that CD36 expression was observed in immune and stroma cells while CD47 was ubiquitously expressed by most types of cells. In immune clusters (bottom), CD36 was mainly expressed by myeloid cells, but it was also expressed by some CD8⁺ T cells.

Following the reviewer's comment, we performed in vitro experiments on T cells isolated from WT, *Cd47*^{-/-} and *Cd36*^{-/-} mice to examine the differences in the activation of downstream signaling in the presence or absence of the ligand (THBS1) (new Fig. 4h and 4i). Notably, the loss of CD47 significantly blocked the suppressive effect of THBS1 on the activation of CD8⁺ T cells (new Fig. 4h), which is in agreement with a previous report demonstrating that THBS1-CD47 signaling directly inactivates T cells by suppressing TCR signaling (Li Z et al., *J Immunol*, 2001). With regard to CD36, we first checked the changes in macrophages of orthotopic MTO tumors because a previous report demonstrated that THBS1 acts on macrophages to suppress lung inflammation by binding to macrophage CD36, which further inhibits T cell function by secreting IL-10 (Akdis CA et al., *Immunology*, 2001). This is consistent with the fact that myeloid cells are the primary cells that express CD36. However, there was no change in IL10 expression in orthotopic tumors

in either *Thbs1*^{-/-} or *Cd36*^{-/-} mice compared to that in the control (new Fig. 4f and Supplementary Fig. 2r). Furthermore, there was no distinct difference in TAM population or expression of TAM-related genes between WT and *Thbs1*^{-/-} mouse tumors (Supplementary Fig. 2p, 3e, 3f). These data suggest that macrophage/IL10 is not involved in the immunosuppressive activity induced by the THBS1-CD36 axis in this model. Notably, the THBS1-induced suppressive effect on T cell activation was also impaired in CD36-deficient CD8⁺ T cells (Fig. 4i). These data indicate that both CD47 and CD36 are involved in the immunosuppressive function of THBS1, wherein THBS1 directly suppresses CD8⁺ T cell activity.

As suggested by the reviewer, we performed a vascularization assay. Consistent with our in vivo data (new Fig. 2d and Supplementary Fig. 2m) and previous reports (Bagavandoss P et al., *BBRC*, 1990; Dawson DW et al., *J Cell Biol*, 1997; Jiménez B et al., *Nat Med*, 2000), the addition of THBS1 blocked the vascular formation of endothelial cells (C166) in vitro. Notably, the simultaneous use of a CD36 inhibitor (but not a CD47 inhibitor) blocked the inhibitory effect of THBS1 on angiogenesis (Fig. 4j). We understand that neither *Cd47*^{-/-} nor *Cd36*^{-/-} mice were able to completely phenocopy *Thbs1*^{-/-} mice because both CD47 and CD36 are multifaceted and function differently depending on their binding partners. Other THBS1 receptors or ligands for CD47 and CD36 may explain these effects through parallel rather than direct mechanisms. A recent study demonstrated that fatty acids suppress antitumor immune activity by binding to CD36 on CD8⁺ T cells (Ma, X et al., *Cell Metab*. 2021). However, our new findings that the loss of either CD47 or CD36 impairs THBS1-induced immunosuppression in CD8⁺ T cells provide a mechanistic insight into how THBS1 contributes to the formation of the immunosuppressive and angiogenic TME. We have included these new data in the revised manuscript.

5. The authors mention THBS1 was increased in CMS4 and CCS3 CRC subtypes. Colorectal Cancer Subtypes (CCS) classification is not widely used. Can the authors state the value of using this classification system in this report? They should update their CMS classification (I Jonito et al., *Nature Genetics* 54:963-975, 2022).

Thank you for this helpful suggestion. As the reviewer has suggested, the CCS classification has not been widely used because it was integrated into the CMS classification (Guinney J et al., *Nat Med*, 2015). In addition, considering that the dissection of cellular and molecular characteristics of individual cell types are necessary to precisely understand the biology of each classified CRC, the updated CMS classification reported by Jonito et al., who used scRNA-seq for their classification, should be included in our analysis. Therefore, we replaced the CCS classification with the updated CMS classification (referred to as the IMF classification). Importantly, we discovered that i3-MSS-F CRC, a subtype with the poorest prognosis, exhibited the highest THBS1 expression, followed by i2-MSS-F CRC. These data further support our hypothesis that THBS1 plays a pivotal role in the progression of aggressive CRC with a mesenchymal phenotype. The new data are shown in Fig. 1d.

6. The authors are encouraged to examine whether THBS1 is increased in MSS vs MSI-H CRC. Our bioinformatics analyses of TCGA samples showed that MSS CRC exhibited higher THBS1 expression than MSI-H CRC (Fig. 7a). Since MSS CRC is a mixed cohort, including both stroma-active (THBS1 high) and stroma-inactive CRC (THBS1 low), the result becomes more evident when we sub-group the samples following the IMF classification (see the above comment #5).

THBS1 expression in i3-MSS-F CRC was higher than that in MSI-H CRC (Fig. 1d). These data are consistent with our hypothesis that THBS1 induces an immunosuppressive TME in CRC with a mesenchymal phenotype. This was further supplemented by the analyses of our own TMA cohorts that demonstrated the association of THBS1-high CRC with a low frequency of MSI-H, and multivariable logistic regression analysis, which showed that high THBS1 expression was independently correlated with MSS (new Fig. 7b and Supplementary Fig. 1m).

Reviewer #2: with expertise in colorectal cancer, metastasis

We are pleased to note that the reviewer is of the opinion that “the authors are to be recommended on a very extensive array of experimentation, the use of different transgene mice and the numerous innovative approaches.”

However, data supporting the contribution of myeloid cell-derived THBS1 in CRC are limited, and there are some confusing points for explanation or further experimentation”. Thank you for your valuable comment and suggestion. We hope that our extensive revisions and new data will clarify our message and satisfy your concerns.

1. The authors found that THBS1 expression is mainly in the stroma of CRC samples by IHC staining in Fig.1a, while the mRNA expression of THBS1 were analyzed in Fig.1d by using TCGA datasets, in which were mainly tumor tissues. Therefore, the bioinformatic analysis in TCGA is not representative, also for the survival analysis. It is better to use your own TMA array to do the survival analysis.

Thank you for this helpful suggestion. We understand the concern raised by the reviewer; however, we still find it worth using the TCGA dataset in our analyses, particularly as a supplement to the results from our own TMA cohort. Although TCGA samples did not discriminate between tumor and non-tumor cells (including stroma and infiltrating immune cells), the gene expression profile of bulk tumor tissues still reflected the distinctive characteristics of whole tumors (not just tumor cells). For example, the consensus molecular subtyping (CMS) classification, which is based on the mRNA expression of all cell types in bulk tumor tissues, has been widely used to study the role of molecules expressed in the TME. In particular, CMS4 CRC, which is highly desmoplastic and contains a large number of stromal components, exhibits high expression of stroma-related genes. This indicates that gene expression profiles in bulk tissues can reflect the characteristics of stroma-rich, aggressive CRC and that THBS1 may play a role in this. We agree with the reviewer that drawing solid conclusions based solely on analyses of bulk tissues in TCGA is not an adequate approach and that it is essential to reinforce the data by combining it with other approaches, including TMA analyses and single-cell transcriptomes, as we have done in this study. To respond to this reviewer, we have moved TCGA analysis data (Fig. 1d in the previous version) to Supplementary Fig. 1b and 1c and have also included the analyses using the updated CMS classification (new Fig. 1d; I Jonito et al., *Nat Genet*, 2022), in which gene expression analyses at single-cell levels were applied to TCGA samples, which may further strengthen the data presented here. Furthermore, we improved the analysis of our TMA samples for multiple prognostic factors and survival (new Fig. 1m, 1n, new Supplementary Fig. 1l, 1m and Supplementary Tables 1–4).

2. Fig.1i, the authors revealed the relationship between THBS1 expression and masson/CD14 by showing IHC staining. For the same CRC samples, it is better to do the staining on the consecutive sections, which will be more representative.

Following this comment, we performed histological analyses with IHC using consecutive sections to strengthen our conclusion (new Fig. 1g and Supplementary Fig. 1j). We observed a correlation between THBS1 expression and the expression of other markers (Masson's trichrome, CD14, CD68, and CD11b).

3. The study focus on the TME-derived THBS1 in CRC. The CRC cells can also produce THBS1, the role of tumor cell-derived THBS1 should be investigated and excluded. How is the THBS1 expression in tumor cells? The expression of THBS1 in organoid and MC38 cells should be detected. Whether the tumor cell-derived THBS1 can contribute CRC progression? Thank you for the suggestion. To compare THBS1 expression levels in the tumor epithelium and stroma, we performed RNAscope analysis on both human CRC samples and mouse MTO orthotopic tumors. RNAscope analysis demonstrated that THBS1 was strongly expressed by infiltrating cells in the tumor stroma and was not detectable in the tumor cells (new Fig. 1b and Supplementary Fig. 2c). Although not detectable by IHC and RNAscope, quantitative RT-PCR analyses revealed a very low level of THBS1 expression in (EPCAM⁺) MTO cells (new Supplementary Fig. 8e; approximately 5% of that in BM-derived cells). MC38 cells did not exhibit significant THBS1 expression (new Supplementary Fig. 8e). To investigate the potential contribution of low levels of THBS1 produced by MTO cells, we generated THBS1 knockdown MTO (*Thbs1* KD) and compared them with shNT MTO (scramble). *Thbs1* KD MTO showed no change in growth or metastatic potential in the splenic injection experiment (new Supplementary Fig. 4b–g). These data clearly support our notion that TME-derived THBS1 contributes to the progression of CRC with a mesenchymal phenotype.

4. For the bioluminescence imaging, such as Fig. 2e, 5e, 6i, 6r, 7d, 7i, 7q, the authors showed the primary lesion (white), lymph node (orange), and liver (yellow) metastasis. How did the authors define the signaling are in the lymph node or liver? It is better to combine CT with IVIS imaging to better define the lymph node and liver. Or to isolate the lymph node and liver, to do the ex vivo imaging, to check the luciferase signaling from lymph node and liver. The bioluminescence signal results were verified by macroscopic and histological findings to confirm whether the positive signals represented lymph nodes or liver metastases. In addition, ex vivo IVIS imaging confirmed metastasis to the liver and lymph nodes and demonstrated positive signals (new Supplementary Fig. 4a). However, we did not always confirm this using ex vivo imaging in the experiments in which we applied bioluminescence imaging. Therefore, we have avoided discriminating between lymph nodes and liver metastases on imaging in the revised manuscript. We have now denoted it as just “distant metastasis” instead of “lymph node metastasis” or “liver metastasis.”

5. Fig. 3j, the authors performed the stimulation experiment on CD3+T cells with and without recombinant THBS1 treatment (25ug/ml). How did the authors define the concentration of recombinant THBS1? Is this concentration-dependent effect? Why using the CD3+T cells? It is better to isolate the CD8+T cells to perform this experiment. For the activation of T cells, normally both anti-CD3 and anti-CD28 were used. Why did the authors use anti-CD3 only, which is not enough to activate T cells?

Thank you for the suggestion. For the experiment indicated (Fig. 3j in the previous manuscript), we referred to a previous report that demonstrated an inhibitory effect of THBS1 on T cell activation (Li Z et al., *J Immunol*, 2001), in which the authors used anti-CD3 to stimulate T cells and the concentration of recombinant THBS1 was set in the range of 0 to 62.5 ug/ml. They found the inhibitory effect of THBS1 at a concentration above 12.5 ug/ml and used mainly 30 ug/ml to

stimulate Jurkat T cells. We tested 12.5, 25, and 62.5 ug/ml and found that 25 ug/ml of THBS1 showed a consistent inhibitory effect on T cells isolated from mice. We agree with the reviewer that it would be better to use anti-CD3 and anti-CD28 as stimuli and analyze CD8⁺ T cells in this experiment. We analyzed CD8⁺ T cells sorted by flow cytometry and found that THBS1 treatment suppressed CD8⁺ T cell activation, which was mediated by anti-CD3 and anti-CD28 antibodies in a concentration-dependent manner (new Fig. 2m–o).

In addition, the protein expression of IFN-g and CTLA4 should be checked by flow cytometry, which makes more sense.

Following this comment, we used flow cytometry to examine the expression of IFN γ , CTLA4, and other molecules. The new data are included in new Fig. 2n and 2o.

6. Fig.6f, the magnified region should be marked in the figure, to make it more visible and obvious.

The magnified region of the corresponding figure has been marked in the revised version of the manuscript (new Fig. 6f).

7. Sup Fig.4c, the full WB blot should be provided, and the molecular weight should be marked.

Full WB has been included and the molecular weight is marked in the revised version of the manuscript (new Supplementary Fig. 6c).

8. Sup Fig5a, Sup Fig.6h, the scale bars for THBS1 and CD31 staining were different.

We corrected the scale bars in the corresponding figures (new Supplementary Fig. 7a and 8k).

Reviewer #3: with expertise in bioinformatics, immunology

We are pleased to see that this reviewer appreciates the impact of our study (“this manuscript reveals a clear mechanism in a frequent, aggressive cancer for recruitment of bone marrow derived myeloid populations, their contribution to generating an immunosuppressive microenvironment, and their relationship with metastasis”) and finds that “This could be appropriate for Nature Communications”.

However, “there are a few major issues (see below) that need to be addressed before this is clear”.

We would like to thank the reviewers for their insightful suggestions, which helped us improve our study. We have revised the manuscript and incorporated new data in response to the reviewers’ comments. We hope that the reviewer will find our revisions satisfactory and suitable for publication.

Major comments

1. The correlative analyses in Figure 1 and Supplementary Figure 1 connecting THBS1 expression with eg prognosis, CRC subtype, gene expression profiles, etc need to more clearly demonstrate how much THBS1 independently drives these patterns versus just being a marker of the CMS4 subtype.

Thank you for this helpful suggestion. We performed univariate and multivariate analyses to investigate the association between THBS1 expression and prognosis (overall mortality, cancer-related death, and recurrence), CRC subtype (clinical stage (I–III), invasiveness (T1–4), primary tumor location, lymph node metastasis (N0–3), lymphatic invasion, venous invasion, undifferentiated/mucinous histology, and MSI/MSS), and gene expression profile (CMS4 or not). Univariate analysis (new Supplementary Fig. 1l) demonstrated a significant association between THBS1 expression and aggressive characteristics (advanced clinical stage, lymph node metastasis, positive undifferentiated histology, and MSS) and a poorer prognosis (high incidence of recurrence, increased mortality, and cancer-related death). Importantly, multivariate logistic regression analyses (new Supplementary Fig. 1m and Supplementary Tables 1–4) revealed that higher THBS1 expression independently correlated with recurrence (OR = 2.48 (1.19–5.18), $P = 0.046$), lymph node met positive (OR = 2.32 (1.27–4.22), $P = 0.019$), undifferentiated histology (OR = 5.72 (1.44–22.7), $P = 0.022$), and MSS (OR = 2.94 (1.50–5.74), $P = 0.004$). Regarding the association with patient survival, multivariate logistic regression analyses defined high THBS1 expression as an independent predictor of overall mortality (OR = 2.69 (1.21–5.99), $P = 0.03$) and cancer-related death (OR = 4.29 (1.52–12.0), $P = 0.01$), whereas multivariate Cox regression analysis revealed no significant difference (HR = 1.19 (0.85–1.67), $P = 0.291$). The TMA samples comprised only surgically resected samples (stages I–III) and did not include stage IV CRC with metastases, in which THBS1 plays an important role, which might have hindered the acquisition of more obvious results. However, the Kaplan-Meier curves revealed a correlation between higher THBS1 expression and lower overall, disease-specific, and recurrence-free survival rates (new Fig. 1m and 1n). Although there were some limitations to our analyses, these results clearly suggest a possible role of THBS1 in independently driving the aggressive patterns of mesenchymal CRC.

Additionally, much more justification/information is needed for the splitting of samples by tiers of THBS1 expression. In Fig 1 d-f, why is it not just by median ?

We downloaded the survival data from the Human Protein Atlas (<https://www.proteinatlas.org/humanproteome/pathology/method>) and Prognoscan (<https://www.ncbi.nlm.nih.gov/pmc/articles/PMC2689870/>), which are widely used web-based resources for survival analyses (we apologize for the lack of clarity in the previous manuscript). We used the cut-off automatically set by these websites based on the FPKM value of each gene. The data from these websites have been well-accepted and published in several papers (e.g., Goto Y et al., *Nat Commun.* 2015 6:6153; Liu Y et al., *Cancer Discov.* 2013 3:870-9). However, in response to the concern raised by this reviewer, we have reanalyzed the survival of these datasets by splitting samples just by the median to obtain the same *n* for both THBS1 high and low groups. Using this new approach, we found that high THBS1 expression was associated with a poor prognosis, and we have replaced the survival data in the revised manuscript (new Fig. 1f and Supplementary Fig.1f).

In Fig 1 o, why are intermediate and high grouped together ?

In response to the reviewer's concern, we have separated the THBS1 high and THBS1 intermediate groups and show the data of all three groups as high (n=107), intermediate (n=122), and low (n=102) in the revised figure. The log-rank test revealed significant differences in OS, DSS, and DFS. Please note that we have increased the number of TMA samples in the revised manuscript. These new data have been included in new Fig. 1m and 1n.

2. The timing in Figure 2 is not clear. If tumor is implanted at week 8, what does week 0 refer to ? This is particularly important for interpreting the survival analysis in panel h.

In Figure 2 of the previous manuscript, "week 0" refers to the birth of the mice used for the experiments. In the revised manuscript, we set "week 0" as the timing of tumor cell implantation in all figures to avoid any confusion.

Given the strong effects of *Thbs1*^{-/-} early in Figure 2, especially e-g, the KM curves in panel h are very surprising. Can the authors explain why there is just a modest delay and then all the *Thbs1*^{-/-} die as rapidly as the WT ? Are they dying of the disease in a different way ?

Although *Thbs1*^{-/-} mice showed a clear reduction in metastases and an increase in apoptotic cells in tumors, there was sustained growth of the primary tumors. In the revised manuscript, we have extended our analyses to include a larger number of mice. We observed longer survival in *Thbs1*^{-/-} mice implanted with MTO in the rectum (new Fig. 3d). However, growing primary tumors eventually killed most of the *Thbs1*^{-/-} mice, although *Thbs1*^{-/-} mice did not show distant metastases or peritoneal dissemination, which frequently occurred and killed WT mice in the experiment.

3. In all the scRNAseq analyses, more extensive explanation/justification must be provided for all cluster annotations (eg with heatmaps, differential gene expression analysis, marker genes, or other approaches).

As requested, we have provided a more extensive explanation by including heat maps and marker genes in the revised manuscript for all scRNA-seq analyses (new Fig. 2, Fig. 5, Supplementary Fig.3 and Methods).

4. The results in Fig 7 on THBS1 inhibition in combination with other treatments do not demonstrate a compelling synergistic effect and therefore should not be highlighted in the title and abstract.

We agree with this observation. As suggested by the reviewer, the most compelling result of *Thbs1*^{-/-} mice in this study is the suppression of metastasis. We toned down our claim regarding the synergistic effects of THBS1 inhibition in combination with other treatments in the revised manuscript, including the title and abstract.

Relatedly, survival analysis and tumor growth kinetics are standard analyses in these types of in vivo tumor treatment models – why are they not included here ?

The evidence connecting THBS1 to metastatic activity (eg Fig 2g, 4e, 5f, 6h and to a lesser extent 6s), conversely, is much more compelling and perhaps does warrant highlighting early and clearly.

In response to the comments of the reviewer, the tumor growth kinetics of all treatment experiments have been included in the revised manuscript (new Fig. 7f and 7n). Survival data were not available for these experiments because the treatment duration was too short to analyze its effect on survival. The treatment duration was set by referring to a previously published study in which similar treatments were performed (Martinez-Ordonez et al., *Cancer Cell*, 2023). We also observed that many mice, including both WT and *Thbs1*^{-/-} mice, became sick and some died when treated with anti-PD1 or anti-VEGFR antibodies for more than 4 weeks, probably due to the anaphylaxis induced by those antibodies. Repeated (4–5 times) administration of anti-PD1 and anti-PDL1 monoclonal antibodies was reported to induce fatal xenogeneic hypersensitivity reactions (IgG mediated pathway of anaphylaxis) specifically in a murine cancer model (Mall et al., *Oncoimmunology*, 2016). This phenomenon is not restricted to anti-PD1/PDL1 antibodies, and other monoclonal antibodies also induce IgG mediated anaphylaxis when used in mice (Murphy et al., *Blood*, 2014). We anticipate that this fatal adverse effect would make it difficult for us to interpret the survival data even if we extend the treatment duration.

As suggested by the editor and the reviewer(s), the most compelling result of *Thbs1*^{-/-} mice is suppressed metastasis. Therefore, considering the limited space of this study, we focused more on investigating the role of THBS1 in metastatic activity than the synergistic effect of combination treatment. We have extensively analyzed the differences in the immune TME between liver metastases and primary tumors and have highlighted how THBS1 inhibition affects the immune TME of liver metastases compared to that of primary tumors (new Fig. 3 and Supplementary Fig. 5) instead of expanding our analyses on the synergistic effect of combination treatment. Using this approach, we believe that we have significantly improved the manuscript.

Minor comments

1. (line 108) “These results suggest that the stroma-infiltrating cells are the primary source of THBS1 and demonstrate a strong association between high THBS1 level, aggressive phenotype,

and poor prognosis in human CRCs” and Fig. 1c. The expression of THBS1 appears higher in the normal colon than in the tumor epithelium. Is it significantly different? Can you provide a possible explanation? Additionally, the violin plots should be replaced by individual data points since n is relatively low.

Yes, there was a significant difference between the normal colon and the tumor epithelium. However, normal colon samples include both the epithelium and stroma, which complicates the interpretation of the results. Furthermore, n of the “normal colon” group was just 8, which was lower than other groups. Therefore, we have removed the “normal colon” group from the analysis in the revised manuscript. In addition, as requested by the reviewer, the violin plots have been replaced with individual data points (new Fig. 1c).

Finally, at this point in the manuscript it is demonstrated that THBS1 expression comes from the stroma, but not that it is from infiltrating cells.

According to this comment, we have changed our description from “stroma-infiltrating cells” to “stroma” in the corresponding part of the revised manuscript.

2. Fig. S 1d, e

Why are there different n for the THBS1 high and low categories between these two plots from the same dataset/cohort?

As explained above (major comment #1), the splitting of the samples was based on a web-based resource in the previous version of the manuscript. We have now split the data by the median to have the same n for comparing the THBS1 high and low groups (new Supplementary Fig. 1f).

3. Fig. 2h,i,j

Can these be re-sized to avoid having the awkward “x10” above the vertical axes in the plot at far right? It is currently confusing to compare with results of similar experiment in panel c. We resized these graphs to avoid confusion (new Fig. 3f and 3h).

4. Fig. 3c

Although the statistical test (Mann-Whitney, aka Wilcoxon) indicates the differential expression is significant, the violin plots are not convincing. Perhaps another type of visualization would be more useful? Ideally, validation by FACS would be even better.

As substantial number of samples with “zero” expression were included in both groups for these genes, the results seemed unconvincing, as the reviewer suggested. Therefore, we have removed these data from the revised manuscript. In response to this reviewer’s comment, we performed FACS analyses to validate the results and found a reduction in CTLA4 but not in PDCD1 (new Supplementary Fig. 3h). Consistent with this result, double-immunostaining demonstrated a reduction of CTLA4⁺ exhausted CD8⁺ T cells and increase in TCF7⁺ active CD8⁺ T cells in *Thbs1*^{-/-} tumors (new Fig. 2p). This was further supported by in vitro experiments, which demonstrated that THBS1 directly increased the proportion of CTLA4⁺CD8⁺ cells in a dose-dependent manner (new Fig. 2o).

5. (line 197) and Fig. 3k

Please provide a citation for these genes as IFN γ target/marker genes. Similar for all signatures/gene sets analyzed in the paper.

We have provided a citation for the IFN gamma signature genes in the Methods section, which were obtained from the Hallmark Collection of MSigDB.

6. (line 231-232) and Figure 4

The phenocopies between *Thbs1*^{-/-}, *CD36*^{-/-}, and *CD47*^{-/-} are clear. However, attributing this directly to interactions between THBS1 and these proteins and referring to “THBS1-CD47 and THBS1-CD36 axes” seems premature without double mutants since, at least a priori, other receptors for THBS1 or other ligands for CD47 and CD36 could explain these effects through parallel rather than direct mechanisms.

Double immunofluorescence confirmed that CD8⁺ T cells infiltrating orthotopic MTO tumors expressed both CD47 and CD36. In vitro experiments analyzing CD8⁺ T cells from WT, *CD47*^{-/-}, or *CD36*^{-/-} mice revealed that loss of either CD47 or CD36 significantly impaired the suppressive effect of THBS1 on activation of CD8⁺ T cells (new Fig. 4h and 4i). These data suggest that both CD47 and CD36 are involved in the immunosuppressive function of THBS1 to directly suppress CD8⁺ T cell activity. However, we agree that we cannot exclude the possibility that other receptors for THBS1 or ligands for CD47 and CD36 could explain these effects through parallel mechanisms. We revised the description in the revised manuscript to avoid the overstatement.

7. Fig. 5i-k

These same genes should also be shown in the mouse scRNAseq data.

As requested, we have provided the plots of mouse scRNA-seq data of orthotopic MTO tumors (new Fig. 5h–j and Supplementary Fig. 3c–d,7c). Please note that marker genes to define human monocytes and mouse monocytes are different (Geissmann F, et al. *Immunity*, 2003;19(1):71-82; Ziegler-Heitbrock L, et al. *Blood*. 2010;116(16):e74-80; Zilionis R, et al. *Immunity*. 2019;50(5):1317-1334 e10). In our analysis, monocyte-like cells showed a gene expression pattern of classical monocytes (CD14⁺, CD16⁻ in human CRC; Ly6C^{hi}, CCR2⁺, CX₃CR1^{int} in mouse MTO tumors), which is in good agreement with previous studies (Movahedi K et al. *Cancer Res*. 2010;70(14):5728-5739, Franklin R.A. et al. *Science*. 2014;344(6186):921-925).

8. Fig. S 5b

As with major comment 1, provide these plots divided by CRC categories.

As requested, we have provided plots for each CRC category (Fig. 5p).

9. Fig. 6b

Please provide the total number of cells in each condition in addition to the percentage to rule out eg a decrease in lymphocytes.

We have included the total number of cells in the figure (Fig. 6b).

10. (lines 303-304) Fig. 6l and Supplementary Figure 6g

Please provide statistical tests that these genes are specifically upregulated in mesenchymal CMS4 CRCs.

In response to the comments from the reviewer, we have reanalyzed the TCGA dataset and provided the data of statistical tests for genes that were specifically upregulated in mesenchymal CMS4 CRCs (new Supplementary Fig. 8i).

11. (lines 303, 307) Fig. 6m and Supplementary Figure 6g

It is confusing that CCL2 is high in CMS4 from TCGA but then lower in the MTO compared to MC38. Can the authors provide any hypotheses?

Thank you for this comment. The importance of CCL2 in MC38-derived tumor models has been reported in several papers (e.g., Zhao L et al., *Hepatology*. 2013;57(2):829-39; Tu W et al., *Cell Death Dis.* 2021 Sep 27;12(10):882). Together with our bioinformatics data showing that CCL2 is upregulated in both CMS1 and CMS4 compared to the other groups (CMS2 and CMS3; new Supplementary Fig. 8h), we anticipated that CCL2 expression would be potentially induced by a certain inflammatory condition, regardless of whether it is an antitumor or pro-tumor immune response. We observed that CCL2 expression was downregulated in MTO-derived tumors. Although MTO is an excellent model of CMS4 CRC (Tauriello DVF, *Nature*, 2017), it may represent a subtype of CMS4 CRC in which CCL2 may not be involved. Recent updates on the CMS classification (IMF classification; I Jonito et al., *Nature Genetics* 54:963-975, 2022) categorized CMS4 CRC into two distinct groups: i3-MSS-F CRC and i2-MSS-F CRC, suggesting that CMS4 is still a heterogeneous group. This may indicate that not all CMS4 showed high CCL2 expression, as MTO did not. To robustly examine the role of CCL2 in CMS4 CRC, a panel of CMS4 CRC organoids (ideally derived from human CMS4 CRCs) would be required for the orthotopic model, which will be achieved in future studies. Due to the limited space of the paper, we were unable to discuss this in depth; however, we have emphasized that CMS4 is not a coherent group and may create a different array of chemokines depending on its epithelial subtype in the revised manuscript.

12. Fig. 7

Why does the data for the WT and Thbs1-/- control conditions appear identical in panels f and s, but different from same conditions in panel n?

We apologize for the inappropriate presentation of data. We performed a treatment experiment with anti-PD1 and anti-VEGFR antibodies as one experiment and compared it with one control condition (i.e., the experiment was performed with three conditions: control vs. anti-PD1 ab-treated vs. anti-VEGFR ab-treated). We have corrected the description in the text and presented the figures.

All violin plots in this figure should be replaced by individual data points.

As requested by the reviewer, we have replaced all violin plots with by individual data points in the figure in the revised manuscript.

The evidence that combination therapy is synergistic or even additive here, outside of potentially panels e and r, is not compelling.

We agree with the reviewer's comments. We toned down our claim regarding the synergistic effects of THBS1 inhibition in combination with other treatments in the revised manuscript.

Reviewer #4: with expertise in cancer immunology, myeloid cells

We thank the reviewers for their valuable comments and suggestions. Following the reviewer's comments, we have extensively revised the manuscript to reveal the role of THBS1 in the regulation of the immune TME, with a particular focus on the difference between primary tumor and metastasis, which provides great insight into why THBS1 inhibition primarily suppresses metastasis. We believe that we have addressed all the concerns raised by the reviewer and hope our revisions are found satisfactory and suitable for publication.

1. The authors observed a substantial increase in CD8⁺ T cell infiltration and cleaved caspase-3 (C-Cas3)⁺ apoptotic cells in orthotopic tumors from *Thbs1*^{-/-} mice. Treatment with anti-CD8 antibody effectively depleted CD8⁺ T cells and, concomitantly, decreased apoptotic cells in the primary tumors (Fig. 2i,j), confirming the cytotoxic activity of the infiltrating CD8⁺ T cells. How do the authors explain that there is no significant difference in tumor growth in KO mice? If THBS1 contributes to immunosuppression, why are T cells not able to kill tumors in KO mice?

We thank the reviewer for this valuable comment. Primary tumors in *Thbs1*^{-/-} mice exhibited an increased number of C-Cas3⁺ apoptotic cells and extensive necrotic areas compared to those in WT mice, which was consistent with the substantial increase in CD8⁺ T cells infiltrating the primary tumors. However, the size of the primary tumor was comparable between WT and *Thbs1*^{-/-} mice. Increased angiogenic activity in *Thbs1*^{-/-} mice may explain this controversy in part. However, these data rather indicated that THBS1-mediated immunosuppression was more pronounced in metastatic lesions than in primary tumor sites. Of note, our additional data on comparison of the immune TME revealed that although both primary and metastatic sites in *Thbs1*^{-/-} mice showed increased CD8⁺ T cell infiltration, the degree of increase was greater in metastasis than in primary tumor (new Fig. 3m and Supplementary Fig. 5a), suggesting the stronger induction of antitumor immunity in metastasis by THBS1-deficiency. We propose that this different immune activity between lesions was attributed to the difference of MDSC accumulation in these lesions (new Fig. 3p,q and Supplementary Fig. 5d). Thus, THBS1 inhibition mainly affected the development of metastasis rather than primary lesions. We have described these issues in more detail in our responses to the following comments (please see #2, #3, and #4).

2. *Thbs1*^{-/-} mice showed increased vascular formation, thus I was wondering if there is a change in the myeloid compartment that is known to contribute to the immunosuppressive TME. No changes in immunosuppressive myeloid cells infiltration may explain why T cells are still ineffective in killing tumors in those KO mice.

We performed extensive analyses to study the immune phenotypes of primary tumors and metastases. scRNA-seq suggested that there was a change only in the CD8⁺ T-cell population and not in other populations, including the myeloid compartments, in the immune TME of primary tumors. These results were confirmed by FACS (Supplementary Fig.2p). Further, we have provided data on the M2 macrophage-related gene signature, which represents the immunosuppressive function of the myeloid lineage, and their marker genes showed no significant differences between WT and *Thbs1*^{-/-} mice (new Supplementary Fig. 2r and 3f). Thus, as the

reviewer suggested, while CD8⁺ T cells were increased and more active in the primary tumors of *Thbs1*^{-/-} mice, these changes were still insufficient to reduce the primary tumor size.

3. The authors showed reduced metastatic lesions, therefore it would be critical to compare the immune TME and THBS1 levels in primary tumor vs metastatic lesions. Such differences in immune composition and THBS1 levels may explain why there is reduction in metastasis lesions only.

This is a critical question, and we have made a huge effort to address it (see the new Fig. 3l–s and Supplementary Fig. 5a–d). Following the reviewer's suggestion, we first compared THBS1 levels in primary tumors and metastases of MTO-bearing WT mice. However, THBS1 levels were comparable between the two groups, indicating that THBS1 levels do not explain why metastasis was more strongly suppressed at the primary site in *Thbs1*^{-/-} mice (new Fig. 3l). Notably, a comparison of the immune TME revealed a change in the number of infiltrating CD8⁺ T cells between primary and metastatic sites. Although both sites in *Thbs1*^{-/-} mice showed increased CD8⁺ T cell infiltration, the degree of increase was greater in metastasis than in primary tumors, suggesting a stronger induction of antitumor immunity in metastasis by THBS1-deficiency (new Fig. 3m and Supplementary Fig. 5a). This point has been discussed further in the following section (#4), but it provides a clear explanation as to why the impact of THBS1 deficiency on tumorigenesis was more evident in metastasis. We next sought to figure out the underlying mechanism and found that metastatic tumors contained a significantly smaller number of CD45⁺CD11c⁻CD11b⁺Ly6C⁺Ly6G⁺ PMN-MDSC compared to primary tumors in WT mice while the number of CD45⁺CD11c⁻CD11b⁺Ly6C⁻Ly6G⁻ MO-MDSC was comparable between lesions (new Fig. 3p,q and Supplementary Fig. 5d). This suggests that while both PMN- and MO-MDSC work together to suppress antitumor immune activity in primary tumors, MO-MDSC play a dominant role in metastasis, at least in this model. Together with the fact that the monocyte lineage was a primary source of THBS1 (new Fig. 5) and that THBS1 loss affected the suppressive function of MO-MDSC but not of PMN-MDSCs (see answer #5; new Fig. 3r), these data explain why the impact of THBS1 inhibition was more pronounced on metastasis. We understand that there are limitations to our analysis. The low incidence and small size of metastases in MTO-bearing *Thbs1*^{-/-} mice hampered sufficient sample collection from metastases for most assays, including FACS. However, we believe that we have successfully addressed the fundamental question of why THBS1 loss has a greater impact on metastasis than on the primary site in an adequate manner.

4. The authors posited that increased infiltrating CD8⁺ T cells in the orthotopic primary site may account for the reduction of metastatic burden in MTO-bearing *Thbs1*^{-/-} mice. It is also possible that there is an increase of CD8 T cells in metastatic sites that may help to reduce metastatic burden. Thus, what is the frequency of T cells in metastatic lesions in those KO mice? The authors showed that loss of THBS1 blocks T cell exhaustion and increases stem-like CD8⁺ T cells, what is the phenotype of T cells in primary vs metastasis?

We thank the reviewer for the helpful suggestions. As anticipated, immune phenotyping of the TME in primary and metastatic tumors by IHC/IF revealed that the increase in infiltrating CD8⁺ T cells was significantly and substantially greater in metastases than in primary sites in *Thbs1*^{-/-} mice (new Fig. 3m and Supplementary Fig. 5a). We also performed splenic injection of MTO and found that liver metastasis development was strongly suppressed in *Thbs1*^{-/-} mice in this system

(new Fig. 3i–k). THBS1 loss-mediated antitumor activity in the splenic injection model was abolished when CD8⁺ T cells were depleted (new Fig. 3i–k), further validating the importance of infiltrating CD8⁺ T cells, particularly in metastatic lesions, for the antitumor immune response. Consistent with this notion, while CD8⁺ T cells in both primary tumors and metastases of *Thbs1*^{-/-} mice showed high expression of the stem-like marker TCF7, metastases contained a higher number of TCF7-expressing CD8⁺ T cells than primary sites, indicating that the impact of THBS1 inhibition on preventing CD8⁺ T cell exhaustion was more evident in metastasis (new Fig. 3s and Supplementary Fig. 5f).

5. BM-derived monocyte-like cells are the primary producers of THBS1 in stroma-rich intestinal tumors. THBS1 affects the activation of T cells in vitro, therefore monocyte like cells may suppress T cell activation via the release of THBS1. Since myeloid cells used different mechanisms to suppress T cells, I was wondering if myeloid cell derived THBS1 contributes to the immunosuppressive activity of these cells.

As suggested by the reviewer, our data suggest that THBS1 contributes to the immunosuppressive function of tumor-infiltrating myeloid-derived cells (MDSCs). Our in vitro experiments using recombinant THBS1 demonstrated that THBS1 directly inhibited T cell activation and that loss of receptor (either CD47 or CD36) impaired the THBS1-mediated suppressive activity (new Fig. 2m–o, 4h and 4i). These data clearly suggest THBS1 produced by myeloid cells affects the activation of T cells through binding to its receptors CD47 and CD36. To further clarify the association of THBS1 function and monocytic myeloid-lineage, we performed MDSC assays and investigated whether THBS1 inhibition affects the activity of PMN-MDSCs, MO-MDSC, or both. We isolated PMN- and MO-MDSC from WT and *Thbs1*^{-/-} mice and measured their suppressive effects on CD8⁺ T cell activation induced by treatment with anti-CD3 and anti-CD28 antibodies. While THBS1 loss did not change the suppressive activity of PMN-MDSCs, it significantly impaired that of MO-MDSC (new Fig. 3r), which is consistent with our notion that the monocyte lineage among myeloid cell compartments produces THBS1 and indicates that THBS1 contributes to the immunosuppressive activity of MO-MDSC. We also found that the accumulation of PMN-MDSCs was reduced in metastases compared to that in primary sites, whereas MO-MDSC accumulation was not different between primary and metastatic sites (new Fig. 3p,q and Supplementary Fig. 5d), suggesting that MO-MDSC play an important role in suppressing the antitumor immune response in metastasis, which is consistent with the fact that the impact of THBS1 loss was more pronounced in metastatic lesions.

Moreover, what is the expression of other suppressive markers in WT vs KO mice?

In response to the reviewer's comment, we have examined other immunosuppressive markers by FACS, IHC/IF, and scRNA-seq analyses and found no significant difference in the infiltration of lymphocytes, including Tregs and suppressive myeloid cells (new Fig.2e, 3n–q and Supplementary Fig. 2o–q,3c,3e,5b–d). These results were further reinforced by the data in which the macrophage-related suppressive gene signature and representative suppressive marker expression were not significantly different between WT and *Thbs1*^{-/-} tumors (new Supplementary Fig. 2r and 3f).

6. The authors found the highest *Thbs1* expression in CD45⁺CD11b⁺CXCR4⁺ cells in BM of

MTO-bearing mice, suggesting that THBS1-expressing CD11b⁺ cells are recruited via the action of CXCL12-CXCR4 signaling. The authors suggest that these cells (CD45⁺CD11b⁺CXCR4⁺) are monocyte like cells. However other myeloid cells expressed these markers. Therefore to better understand the myeloid cell composition, the authors should characterize CD45⁺CD11b⁺CXCR4⁺ cells in primary tumors and metastasis sites.

In response to the reviewer's concern, we examined THBS1 expression in CD45⁺CD11b⁺CXCR4⁺ cells from primary tumors and metastasis. FACS analyses revealed that CD45⁺CD11b⁺CXCR4⁺Ly6C⁺CCR2⁺ monocyte-like cells in tumors were the primary population of THBS1-expressing cells at both primary and metastatic sites, compared to other cell populations, including CD45⁺CD11b⁺CXCR4⁺Ly6C⁺CCR2⁻, CD45⁺CD11b⁺CXCR4⁺Ly6C⁻, CD45⁺CD11b⁺CXCR4⁻, CD45⁺CD11b⁻, and CD45⁻ cells. These data were further supported by OPAL-IF, which showed the co-localization of Ly6C, CXCR4, and THBS1 at both primary and metastatic sites. These new data have been included in Fig. 6p,q, and Supplementary Fig. 8j.

7. CD8⁺ T cells in 176 *Thbs1*^{-/-} tumors exhibited a decrease in dysfunctional markers such as *Pdcd1* and *Ctla4*. How do the authors explain that anti-PD1 therapy may work in this model?

Fig. 3c in the previous version of the manuscript showed a statistically significant reduction in the mRNA expression of *Pdcd1* and *Ctla4*. However, as suggested by the other reviewer, these results did not appear convincing because the scRNA-seq data for these genes included substantial number of samples with "zero" expression in both groups. Therefore, we re-evaluated the expression of these markers in *Thbs1*^{-/-} tumors by FACS and found a significant reduction in CTLA4 but not in PDCD1 (new Supplementary Fig. 3h). Based on this, anti-PD1 therapy could be expected to be more effective in *Thbs1*^{-/-} tumors, but the combined effect of THBS1 inhibition and anti-PD1 ab was not compelling, despite significantly reducing the growth rate of primary tumors (new Fig. 7g). This could be because the infiltration of CD8⁺ T cells in the primary tumors of *Thbs1*^{-/-} mice was not sufficient to kill all tumor cells, particularly in primary sites where PMN-MDSCs, which were not affected by THBS1 loss in their suppressive function (new Fig. 3r), could compensate for the impaired function of MO-MDSCs (due to THBS1 loss).

Do the mice survive longer?

Survival data were not available from this experiment, since the duration of treatment was too short to analyze the effect on survival. The treatment duration was set by referring to a previously published study in which similar treatments were performed (Martinez-Ordonez et al., *Cancer Cell*, 2023). We also observed that many mice, including both WT and *Thbs1*^{-/-} mice, became sick and some died when treated with anti-PD1 or anti-VEGFR antibodies for more than 4 weeks, probably due to the anaphylaxis induced by those antibodies. Repeated (4–5 times) administration of anti-PD1 and anti-PDL1 monoclonal antibodies was reported to induce fatal xenogeneic hypersensitivity reactions (IgG mediated pathway of anaphylaxis) specifically in a murine cancer model (Mall et al., *Oncoimmunology*, 2016). This phenomenon is not restricted to anti-PD1/PDL1 antibodies, and other monoclonal antibodies also induce IgG mediated anaphylaxis when used in mice (Murphy et al., *Blood*, 2014). We anticipate that this fatal adverse effect would make it difficult for us to interpret the survival data even if we extend the treatment duration.

Thbs1^{-/-} tumors and combo with anti-PD1 showed similar reduction in mets. However, tumors are smaller in KO mice following treatment with anti-PD1. How do the authors explain that a reduction in primary tumor does not reflect in a significant decrease in number of mets? I have observed a similar trend in fig7m and s.

Because THBS1 inhibition did not reduce PDCD1 expression at the primary site, as previously described (new Supplementary Fig. 3h), it is theoretically possible that a combination of THBS1 inhibition and an anti-PD1 antibody exhibits a synergistic effect in both primary tumors and metastases. However, in regards of metastasis, although the combination tended to reduce the number of both liver and lymph node metastases, the reduction was not statistically significant (new Fig. 7h). This could be because THBS1 inhibition itself strongly suppresses metastasis, making it difficult to observe a clear effect of the combination treatment on metastasis. This could also explain why a clear difference in metastatic burden was not observed with other treatments, including anti-VEGFR antibodies and FOLFOX (new Fig. 7h,7p). These data strongly suggest, as suggested by the editor and the reviewer(s), that the most compelling result of *Thbs1*^{-/-} mice is suppressed metastasis. Therefore, considering the limited space of the paper, we focused more on investigating the role of THBS1 in metastatic activity than on demonstrating the synergistic effect of the combination treatment. In the revised manuscript, we have toned down the claim on this part and extensively improved our analyses to reveal the difference in immune phenotypes between primary and metastatic tumors.

REVIEWERS' COMMENTS

Reviewer #1 (Remarks to the Author):

The authors have addressed my previous concerns.

The manuscript is much improved.

Reviewer #2 (Remarks to the Author):

The authors have addressed all of my concerns, I do not have further concerns.

Reviewer #3 (Remarks to the Author):

The authors have done an excellent job of responding to my comments in the first round, particularly concerning the survival analyses and the relative focus on metastasis vs drug sensitivity. The manuscript is significantly improved and I only have a few minor points. Otherwise I am supportive of publication.

Minor comments:

1. I would suggest explicitly adding the n in every figure panel analyzing cohorts, even if it is redundant (ie the same cohort is analysed in multiple panels). For example, Supplementary Figure 1 panels b, c, h, i, and k. This just makes it easier for readers in my opinion.
2. Page 6, line 128. The reference to "Thbs1-/- tumors" is confusing. If I am not mistaken, the MTO malignant cells in this experiment are Thbs1 proficient, but the recipient mouse is Thsb1 -/-. Clarifying the language here would be helpful. The same issue comes up repeatedly in this paragraph as well as for the CD47 and CD36 -/- mice on page 11.
3. Specifying the concentration of THBS1 in figure 4h would help to compare with figure 2n,o.
4. The ex vivo T cell activation and cytotoxicity studies in Fig 2 m, n, o, and q are very interesting and broadly consistent with the in vivo findings. However, the strong block of activation ex vivo by THBS1 seems at odds with the strong enrichment for exhausted/dysfunctional CD8 T cells (which generally is associated with chronic TCR activation) in THBS1 high tumors in vivo. It would be good to address this in the discussion,

perhaps in relation to the strong, acute activation of the ex vivo protocol compared to the more long-term and likely lower-level TCR triggering by endogenous antigens in vivo.

Reviewer #4 (Remarks to the Author):

The authors addressed all reviewer concerns. No further experiments/clarifications are required

RESPONSES TO REVIEWERS' COMMENTS:

We would like to express our gratitude to all the reviewers for their meticulous review and insightful suggestions, which have greatly contributed to improving the manuscript.

Reviewer #1 (Remarks to the Author): The authors have addressed my previous concerns. The manuscript is much improved.

We are pleased to note that the reviewers have found that our revised manuscript has adequately addressed all the concerns raised. Thank you so much for the reviewers' invaluable suggestions.

Reviewer #2 (Remarks to the Author): The authors have addressed all of my concerns, I do not have further concerns".

We genuinely appreciate the reviewers' positive feedback regarding our additional research. The reviewers' invaluable comments and suggestions have significantly contributed to enhancing the manuscript.

Reviewer #3 (Remarks to the Author): The authors have done an excellent job of responding to my comments in the first round, particularly concerning the survival analyses and the relative focus on metastasis vs drug sensitivity. The manuscript is significantly improved and I only have a few minor points. Otherwise I am supportive of publication.”)

We are pleased to note that this reviewer also acknowledges the enhancements made to our revised manuscript. We have revised the manuscript in response to the reviewers' comments.

Minor comments

1. I would suggest explicitly adding the n in every figure panel analyzing cohorts, even if it is redundant (ie the same cohort is analysed in multiple panels). For example, Supplementary Figure 1 panels b, c, h, i, and k. This just makes it easier for readers in my opinion.

Thank you for this helpful suggestion. We have added the n in every figure panel analyzing cohorts (new Fig. 1d, 1j, 2l, 5n, 5p, 6n, 7a and Supplementary Fig. 1b, 1c, 1g-1i, 1k, 3j).

2. Page 6, line 128. The reference to “Thbs1^{-/-} tumors” is confusing. If I am not mistaken, the MTO malignant cells in this experiment are Thbs1 proficient, but the recipient mouse is Thbs1^{-/-}. Clarifying the language here would be helpful. The same issue comes up repeatedly in this paragraph as well as for the CD47 and CD36^{-/-} mice on page 11.

In response to the reviewer's concern, we have revised the reference from “Thbs1^{-/-} tumors” to “tumors in Thbs1^{-/-} mice” through the whole manuscript to avoid any confusion. The description of “Cd47^{-/-} tumors” and “Cd36^{-/-} tumors” has been revised in the same way.

3. Specifying the concentration of THBS1 in figure 4h would help to compare with figure 2n,o. As requested, we have provided the information about concentration of THBS1 in Fig. 4h.

4. The ex vivo T cell activation and cytotoxicity studies in Fig 2 m, n, o, and q are very interesting and broadly consistent with the in vivo findings. However, the strong block of activation ex vivo by THBS1 seems at odds with the strong enrichment for exhausted/dysfunctional CD8 T cells (which generally is associated with chronic TCR activation) in THBS1 high tumors in vivo. It would be good to address this in the discussion, perhaps in relation to the strong, acute activation of the ex vivo protocol compared to the more long-term and likely lower-level TCR triggering by endogenous antigens in vivo.

Thank you for pointing this out. As suggested by the reviewer, because of a difference in the experimental models, THBS1 exerts its suppressive effect on CD8⁺ T cell activity in a different way between ex vivo/in vitro (strong block of activation) and in vivo conditions (induce exhaustion/dysfunction). We have included the discussion about this issue in the revised manuscript.

Reviewer #4 (Remarks to the Author): The authors addressed all reviewer concerns. No further experiments/clarifications are required.

Thank you very much for your valuable comments and suggestions for enhancing our manuscript.